# Comprehensive Analysis of Bioactive Compounds, Functional Properties, and Applications of Broccoli By-Products

**DOI:** 10.3390/foods13233918

**Published:** 2024-12-04

**Authors:** Iris Gudiño, Rocío Casquete, Alberto Martín, Yuanfeng Wu, María José Benito

**Affiliations:** 1Nutrición y Bromatología, Escuela de Ingenierías Agrarias, Universidad de Extremadura, Avd. Adolfo Suárez s/n, 06007 Badajoz, Spain; igudino@unex.es (I.G.); amartin@unex.es (A.M.); mjbenito@unex.es (M.J.B.); 2Instituto Universitario de Investigación en Recursos Agrarios (INURA), Universidad de Extremadura, Avd. de la Investigación, 06006 Badajoz, Spain; 3School of Biological and Chemical Engineering, Zhejiang University of Science and Technology, Hangzhou 310023, China; wuyuanfeng@zju.edu.cn

**Keywords:** broccoli, by-products, bioactive compounds, extraction methods, beneficial effects, applications

## Abstract

Broccoli by-products, traditionally considered inedible, possess a comprehensive nutritional and functional profile. These by-products are abundant in glucosinolates, particularly glucoraphanin, and sulforaphane, an isothiocyanate renowned for its potent antioxidant and anticarcinogenic properties. Broccoli leaves are a significant source of phenolic compounds, including kaempferol and quercetin, as well as pigments, vitamins, and essential minerals. Additionally, they contain proteins, essential amino acids, lipids, and carbohydrates, with the leaves exhibiting the highest protein content among the by-products. Processing techniques such as ultrasound-assisted extraction and freeze-drying are crucial for maximizing the concentration and efficacy of these bioactive compounds. Advanced analytical methods, such as high-performance liquid chromatography–mass spectrometry (HPLC-MS), have enabled precise characterization of these bioactives. Broccoli by-products have diverse applications in the food industry, enhancing the nutritional quality of food products and serving as natural substitutes for synthetic additives. Their antioxidant, antimicrobial, and anti-inflammatory properties not only contribute to health promotion but also support sustainability by reducing agricultural waste and promoting a circular economy, thereby underscoring the value of these often underutilized components.

## 1. Introduction

Broccoli (*Brassica oleracea* var. *italica*) is a member of the cruciferous family (*Brassicaceae*), which also includes vegetables such as cauliflower, brussels sprouts, and kale. The *Brassicaceae* family encompasses up to 390 genera and approximately 3500 species, predominantly found in the Mediterranean region and Central Asia [1]. Among these, the genus Brassica is particularly significant, comprising species of substantial economic and nutritional importance. *Brassica oleracea*, in particular, includes a diverse array of vegetables, with broccoli being one of the most notable.

The Brassica genus constitutes a significant component of the global human diet. Its consumption is particularly prominent in countries such as China, Japan, India, and various European nations, where it is considered a dietary staple [2]. Global production of Brassica species has seen a notable increase in recent years, reaching 2.2 million tonnes between 2015 and 2022. The majority of this production is concentrated in China and India, followed by the United States, Spain, Italy, and Mexico [3]. Broccoli, a member of this genus, is cultivated in a diverse range of climates due to its resilience, facilitating its global cultivation. This adaptability has also enabled the development of various broccoli cultivars.

Broccoli is a vegetable abundant in bioactive compounds, including glucosinolates and their degradation products, polyphenols (such as flavonoids and hydroxycinnamic acids), vitamins (ascorbic acid), minerals (manganese, potassium, and selenium), and carbohydrates (dietary fibers) [4,5]. These bioactive compounds exhibit significant antioxidant, anti-inflammatory, and anticancer activities [6,7]. Despite its numerous beneficial properties, broccoli cultivation primarily targets food consumption, with the edible portion comprising only 15% of the entire plant. This portion includes the florets, often accompanied by tender stems and sprouts. The production of broccoli generates a substantial amount of by-products, such as leaves and stems, which are typically discarded as they are considered non-usable or commercially underutilized [8,9]. However, these by-products offer considerable health benefits due to their richness in a wide array of valuable bioactive compounds, similar to the edible parts. Studies have shown that by-products like stems and leaves contain high levels of phenolic compounds, glucosinolates, vitamins, and fiber [10,11,12]. The analysis of compounds present in broccoli necessitates various extraction and analysis methods. Commonly used extraction methods include solid-phase extraction (SPE) and liquid–liquid extraction (LLE), as well as ultrasound-assisted extraction (UAE) and supercritical fluid extraction (SFE), which are considered more efficient and environmentally friendly. For compound analysis, chromatographic techniques such as high-performance liquid chromatography (HPLC) are employed to separate and quantify the compounds, often in combination with mass spectrometry (MS) or UV–Vis detection to enhance identification [13,14,15].

The presence of these compounds, primarily phenolic compounds and isothiocyanates (ISTs) such as sulforaphane (SF), is responsible for the antioxidant and antimicrobial properties associated with this vegetable [16]. Broccoli, like many cruciferous vegetables, is renowned for its chemopreventive properties, primarily due to its rich content of compounds like SF and other ISTs. These compounds are notable for their antioxidant effects and their ability to regulate the cell cycle, induce apoptosis, and modulate enzymatic activity [17,18,19]. Consequently, the inclusion of isothiocyanates derived from broccoli reduces the susceptibility to developing chronic diseases such as cancer. Similarly, the influence of isothiocyanates, such as sulforaphane, and flavonoids like quercetin on the activity of immune cells involved in inflammation, such as macrophages and lymphocytes, has been demonstrated [19]. Dietary broccoli supplementation (sulforafane) has been found to improve glucose tolerance, attenuate liver inflammation, and alter microbial composition in mice fed high-fat diets [20]. Broccoli stalks have demonstrated positive effects on lipid parameters, insulin sensitivity, and gut microbiome diversity [20]. The use of both broccoli sprouts and leaves and stems has also been studied for their effects on other conditions such as obesity [21,22,23], as well as diabetes [24,25,26,27].

The extraction of compounds from broccoli responsible for its multiple benefits can transform broccoli cultivation by-products into new sources of bioactive compounds. This would allow the reuse of the inedible parts of the vegetable in the food industry as functional or nutraceutical products, promoting the valorization of broccoli agro-industrial waste. This review aims to compile relevant literature, mostly from the last ten years, focusing on the detailed analysis of nutrients and phytochemicals present in various parts of broccoli, as well as in different cultivars. It seeks to evaluate the importance of the choice of extraction and analysis methods in obtaining these bioactive compounds. Recent advances in their beneficial effects are also highlighted, and various applications of broccoli by-products are explored to promote their use as a rich source of compounds with high functional properties.

## 2. Brassica By-Products

The horticultural industry generates substantial by-products at every stage of the production chain, particularly during handling and processing. Any raw material or product that fails to meet quality standards, field waste, and parts not intended for fresh consumption are classified as by-products.

Broccoli production results in significant quantities of by-products, most of which remain underutilized. Since the broccoli head is the most commonly consumed part, the remainder of the plant (leaves, stems, roots, etc.) is typically treated as by-products or waste. Broccoli leaves and stems alone constitute 60 to 75% of the plant’s total biomass and are generally discarded post-harvest [1]. The rest of the plant comprises roots (17%) and florets (15%) [8]. Currently, approximately 73.65 tons per hectare of cultivated broccoli are generated as waste after crop harvesting, with the primary use being reincorporation into the soil as compost for new plantings. However, these by-products can serve as inexpensive sources of bioactive compounds. Broccoli leaves are rich in vitamins E and K, and glucosinolates, often surpassing florets in concentrations of phenolic compounds and chlorophyll, thereby exhibiting higher antioxidant activity, while stems are rich in fiber and glucosinolates [8,11,12,13,14].

The extraction of bioactive compounds from by-products facilitates their incorporation into the production of new products in the food, pharmaceutical, and cosmetic industries. This practice is already being implemented, utilizing broccoli by-products as ingredients in functional foods [28], to enhance the nutritional properties of foods [29], or as natural preservatives to extend the shelf life of foods [30], among other applications. Given the health benefits provided by broccoli beyond basic nutrition due to its wide array of phytochemicals, and the presence of these compounds in its by-products, they can be considered valuable food additives [13,14].

Utilizing these by-products not only offers economic benefits by increasing the value of materials used by other industries but also provides ecological advantages by reducing waste emissions. This approach supports a circular economy by minimizing the input of raw materials and the production of waste. Additionally, it enhances the utilization of plant foods at a time when arable land is decreasing and the global population is increasing.

## 3. Functional Compounds and Extraction Techniques

### 3.1. Phenolic Compounds

Cruciferous vegetables, such as broccoli, are renowned for their high phenolic compound content. In plants, these compounds fulfill various roles, including protection against external agents, cold tolerance, and contributing to color and aroma. Humans cannot synthesize these compounds, thus relying on the consumption of plant-based foods like broccoli to incorporate them into the body.

The total phenolic content (TPC) in mg gallic acid equivalent/100 g dry weight (mg GAE/100 g DW) is highest in the leaves (2435 mg GAE/100 g DW), followed by the inflorescence (1074 mg GAE/100 g DW) and stems (939 mg GAE/100 g DW) [11]. Broccoli sprouts and seeds have the lowest concentrations of phenolic compounds (385.4 and 480.4 mg GAE/100 g DW, respectively) [26]. This ranking is consistent with other studies [8,12], with the phenolic content in leaves only surpassed by discarded florets after processing [10].

The extraction method significantly influences the phenolic compound content analyzed in broccoli by-products. Borja-Marínez et al. [31] reported higher concentrations of these compounds using solvent extraction (MeOH) compared to supercritical fluid extraction (SFE) in mixed samples of leaves and stems. Le et al. [32] applied extraction on leaves and seeds using three different solvents (MeOH, EtOH, and hot water), highlighting the efficacy of alcohol. Solvent extraction, primarily using methanol, is the most commonly employed method for phenolic compound extraction [9,33,34]. However, other solvents such as acetone [35] or ethanol [15] have also been utilized. Martínez et al. [36] applied water extraction along with enzymatic extraction (EE) or ultrasound-assisted extraction (UAE) in fresh broccoli leaf and stem mixtures, emphasizing the effectiveness of water extraction at 100 °C.

The phenolic compound content also depends on the pre-processing of by-product samples. Md Salim et al. [37] found that osmotic dehydration (OD) of samples is more favorable for phenol content than hot air-drying (40–60 °C). However, Villaño et al. [38] achieved higher phenolic compound extraction by pre-drying broccoli stems with air at 80 °C (8.37 mg GAE/g DW) compared to freeze-drying (3.01 mg GAE/g DW). Pre-blanching the plant sample followed by freeze-drying enhances phenolic compound extraction [39].

#### 3.1.1. Flavonoid

Broccoli by-products are a rich and accessible source of phenolic compounds, particularly flavonoids such as quercetin and kaempferol, which exhibit significant antioxidant and anti-inflammatory properties. The total flavonoid content (TFC) in broccoli leaves is 9.93 mg catechin equivalent/g DW (mg CE/g DW), using ethanol (70% *v*/*v*) as the extraction solvent [32]. In another study, the TFC in broccoli sprouts and seeds was 206.9 and 216.9 mg CE/100 g, respectively, using the same solvent [26], demonstrating its effectiveness for extracting these compounds. Solvent extraction (EtOH 80%) assisted by ultrasound (UAE) has also been applied, yielding TFC values of 2.4 mg quercetin equivalent/g (mg QE/g) and 5.4 mg QE/g in stems and discarded florets, respectively [10]. Methanol is another solvent used for flavonoid extraction [9,33].

Thomas et al. [10] quantified the content of k-3-o-sophoroside and q-3-diglucoside-7-glucoside in discarded florets (39.4 and 17.2 mg/kg DW), although these flavonoids were not detected in stems. Berndtsson et al. [40] also identified significant amounts of kaempferols (k) and quercetins (q) in broccoli leaves using UAE with methanol (60%) as the solvent, including k-3-O-caffeoyl-sophoroside-7-O-diglucoside, k-3-O(feruloyl)sophoroside-7-O-diglucoside, q-3-O-(sinapoyl)-sophoroside-7-O-glucoside, and k-3-O-(sinapoyl)-sophoroside-7-O-glucoside, among others. Le et al. [32] determined the quercetin content in leaves and seeds, obtaining 0.97 and 0.56 mg/g of extract (EtOH 70%), respectively. They also measured the myricetin content, another flavonoid, with 2.77 mg/g in leaf extract and 0.31 mg/g in seed extract. This same study also highlighted the presence of another phenolic compound, esculetin, belonging to the coumarin group, with contents of 6.49 and 10.18 mg/g in broccoli leaf and seed extracts, respectively.

#### 3.1.2. Phenolic Acids

Interest in phenolic acids among phenolic compounds has surged due to their antioxidant and antimicrobial properties. Broccoli by-products are a rich source of these compounds, particularly in the leaves (3.6–5.7 mg/g DW) [40]. These compounds are categorized into two main types: benzoic acids, derived from benzoic acid, and hydroxycinnamic acids, derived from cinnamic acid.

Benzoic acids such as vanillic acid and gallic acid have been extracted and identified from broccoli by-products like stems and seeds [32,41], although these compounds were not detected in the leaves [41]. However, Le et al. [32] successfully extracted gallic acid from leaves using a solvent with a higher alcohol percentage (70%) compared to the previous study (50%) [41].

The primary hydroxycinnamic acids present in broccoli include caffeic acid, ferulic acid, chlorogenic acid, neochlorogenic acid, and sinapic acid. The main extraction method for these compounds is solvent extraction, using methanol [10,40,42] or ethanol [32,41,43]. However, Martínez et al. [36] utilized water as the solvent, applying enzymatic extraction (EE) with cellulose, and ultrasound-assisted extraction (UAE). The extraction of caffeoylquinic acid was higher when applying EE in a mixture of broccoli leaves and stems. Conversely, Costa-Pérez et al. [43] investigated the effect of processing broccoli stems on the content of 5-caffeoylquinic acid and di-sinapoyl-diglucoside, derivatives of hydroxycinnamic acids. They extracted a higher concentration of 5-caffeoylquinic acid after freeze-drying the stems (4.30 mg/kg DW), whereas the content of di-sinapoyl-diglucoside was higher after drying them with a descending temperature gradient (14.93 mg/kg DW). These findings suggest that the choice of vegetable processing depends on the target compound. Berndtsson et al. [40] also identified numerous hydroxycinnamic acids such as o-diglucoside and trisinapoyl-diglucoside in broccoli leaves using HPLC-DAD-ESI-MS. Caffeoyl derivatives, caffeoyl-hexose derivatives, and feruloyl-caffeoyl derivatives, among other compounds, were also identified using this analytical method [36].

### 3.2. Organic Acids

Broccoli by-products encompass a diverse array of organic acids, with malic acid, citric acid, and oxalic acid being particularly abundant in the sprouts [42]. The leaves are notably rich in gluconic acid, cinnamic acid, succinic acid, and fumaric acid [11,15,33]. Additionally, the inflorescence is especially rich in gluconic acid with 420.64 arbitrary units of area (UAA), exceeding the levels found in other by-products [11].

Table 1 presents the phenolic compounds and organic acids identified in various parts of broccoli, contingent on the pre-processing techniques applied to the vegetable. It also details the different extraction and analytical methods employed for their determination. 

### 3.3. Glucosinolates and Their Breackdown Products

Glucosinolates (GLSs) are distinctive compounds found in Brassica family vegetables, including broccoli. The predominant method for extracting these compounds involves solvent extraction, primarily using methanol at elevated temperatures (>70 °C) to deactivate the enzyme myrosinase [8,10,39]. Myrosinase hydrolyzes GLSs, producing glucose and aglycone. The aglycone is highly unstable and rearranges to form various hydrolytic products, mainly isothiocyanates (ITCs) and nitriles, along with epithionitriles, thiocyanates, indoles, and oxazolidines [44].

#### 3.3.1. Glucosinolates

The structural variation of GLSs allows these compounds to be categorized into three groups: aliphatic, aromatic, and indolic, based on the side chain structure. The primary aliphatic GLSs in broccoli by-products include glucoraphanin, glucoiberin, glucoerucin, and gluconapin. Glucoraphanin (GLP) is the most prominent, with higher concentrations in the inflorescence (3149.91 UAA) compared to leaves and stems (307.29 and 139.27 UAA) [11]. However, Liu et al. [8] reported higher concentrations in stems than in leaves in the “Green Magic” variety. Wang et al. [45] quantified GLP in seeds (2.5–129.9 μmol/g FW) and Villaño et al. [21] in sprouts (51.08 mg/30 g FW). Additionally, Thomas et al. [10] found higher aliphatic glucosinolate content in discarded floret residues post-processing compared to broccoli stems. Other identified aliphatic glucosinolates include napoleiferin, progoitrin, glucoalyssin, and glucoiberverin [8,10,46], as well as 6-(methylsulfonyl)hexylglucosinolate and 3-butenyl glucosinolate in broccoli leaves [47].

The primary aromatic GLS in broccoli leaves is gluconasturtiin [46], while sinalbin is found in stems [33]. Other GLSs, such as glucobrassicin, methoxyglucobrassicin, 4-methoxyglucobrassicin (4-MGBS), 4-hydroxyglucobrassicin (4-GBS), and neoglucobrassicin (NGBS), belong to the indolic glucosinolates group [8,10,45,46].

The GLS content in broccoli by-products is influenced by the cultivated variety. Ares et al. [46] examined the GLS content in broccoli stems from three varieties (Parthenon, Nubia, and Naxos). The Parthenon variety exhibited higher levels of aliphatic glucosinolates such as GLP, glucoiberin, and glucoerucin, as well as the aromatic gluconasturtiin and the indolic 4-MGBS. Conversely, the “Nubia” variety had higher levels of the indolic glucosinolate glucobrassicin (77–106 μg/g DW).

The pre-processing of broccoli significantly affects glucosinolate extraction. Costa-Pérez et al. [43] analyzed glucosinolate content in broccoli stems (“Parthenon”) following three processing methods: lyophilization, over-drying (40 °C, 72 h), and temperature gradient drying (75–60 °C). Lyophilized stems had higher glucoraphanin (2633.89 mg/kg DW) and methoxyglucobrassicin (449.05 mg/kg DW) content, while oven-dried stems at 40 °C showed higher glucobrassicin (113.27 mg/kg DW) and glucoiberin (566.64 mg/kg DW) extraction. Another study compared lyophilization with hot air-drying at 60 and 80 °C for broccoli stems [38]. Extracts rich in GLP were obtained from stems dried at 80 °C (3.73 mg/g DW), but high temperatures degraded other GLSs like glucobrassicin and methoxyglucobrassicin, which were not detected. Lyophilized stems had high methoxyglucobrassicin content, while air-dried stems at 60 °C were notable for their glucobrassicin content.

#### 3.3.2. Isothyocianates

The formation of products when glucosinolates react with the enzyme myrosinase is contingent upon the glucosinolate’s structure and the biodegradation conditions, including pH, and the presence of proteins and metal ions. Under neutral pH conditions, the hydrolysis predominantly yields volatile active compounds known as isothiocyanates (ITCs) [44]. Among the ITCs identified in broccoli, sulforaphane (SF) is notable, being a degradation product of glucoraphanin (GLP). Lv et al. [9] reported SF contents ranging from 2.5 to 12.1 mg/g FW in fresh broccoli seeds, with an increase from 0 to 11.70 mg/g in sprouts between days 3 and 7 of germination. The SF content was higher when using UAE with ethyl acetate/water as a solvent (17.0 mg/g) [48] compared to high-pressure homogenization (HPH) (2199 μmol/g) [49] in fresh broccoli seeds. Additionally, lyophilization of stems prior to extraction increased the SF content to 1.09 mg/kg DW [43].

Indole-3-carbinol, another ITC, is present in broccoli by-products as a degradation product of glucobrassicin [43]. Abdelhalim et al. [50] identified benzyl isothiocyanate (BITC) in broccoli leaves, formed from the degradation of glucotropaeolin. Other ITCs, such as 4-(methylthio)butyl isothiocyanate (GER-ITC) and 4-(methylsulfinyl)butyl isothiocyanate (GRA-ITC), have also been identified in broccoli seeds [45].

#### 3.3.3. Other Breakdown Products

Nitriles are another kind of degradation product formed from glucosinolates through the action of the enzyme myrosinase. The formation of these compounds occurs under acidic conditions (pH 2–5) [51]. Identified nitriles in broccoli seeds include 5-(methylthio)-pentanenitrile (GER-CN), 5-(methylsulfinyl)pentanenitrile (GRA-CN), 4-methylsulfinylbutanenitrile (GIB-CN), and 4-(methylthio)butanenitrile (GIV-CN) [45].

Table 2 compiles data from various studies on glucosinolates and their degradation products in broccoli by-products, highlighting the influence of different processing conditions and extraction and analysis methods.

### 3.4. Pigments

Broccoli, like many vegetables, contains natural pigments such as chlorophyll and carotenoids. The total chlorophyll content is highest in the leaves (76.42 mg/100 g DW), followed by the inflorescence (35.50 mg/100 g DW), and the stems (13.88 mg/100 g DW) [11], with a higher presence of α-chlorophyll [8]. The extraction yield from different broccoli by-products varies based on the method used, including solvent extraction with acetone [11,37], methanol or ethanol [31,39], solvent mixtures [8], or SFE [31]. Chlorophyll content is also influenced by the processing method. Fresh broccoli has higher chlorophyll levels, which decrease with hot air-drying (40, 60 °C) [37]. However, blanching and lyophilization help preserve more of this pigment [39].

Similarly, carotenoid content is higher in the leaves (1095.0 μg/g DW) compared to the stem (15.6 μg/g DW) [8]. Identified carotenoids include lutein and neoxanthin [8,39], with β-carotene and violaxanthin found only in broccoli leaves [8].

The content of these pigments also varies with the broccoli cultivar. Borja-Martínez et al. [31] reported higher pigment concentrations in the “Naxos” variety compared to “Parthenon”.

### 3.5. Vitamins

Broccoli by-products are a valuable source of essential vitamins. The vitamin C content is highest in the leaves (1.08 mg/g), followed by the stems (0.6 mg/g) [33]. Fresh sprouts exhibit the highest concentration of this vitamin, with 5.45 g/30 g (FW) [21].

The vitamin C content varies not only with the part of the vegetable used but also with the extraction method and prior treatment. Le et al. [32] analyzed the vitamin C content in broccoli leaves and seeds, finding that extraction results varied with the solvent used. The highest vitamin C values were obtained in leaves (2.74 mgAA/g DW) using hot water extraction and in seeds (2.69 mgAA/g DW) using 70% *v*/*v* ethanol. Md Salim et al. [37] demonstrated that the vitamin C content in broccoli stems is influenced by the preparation method, with the highest concentration (4.00 mg/g DW) in fresh stems, followed by osmotic dehydration drying (1.99 mg/g DW). This drying process reduced vitamin C destruction compared to microwave-assisted air-drying (40 and 60 °C), although working with fresh vegetables minimized this loss.

Broccoli also contains vitamin E, with total tocopherol content varying by plant part. The highest levels are found in the leaves (155.0 μg/g DW) compared to the stems (1.97 μg/g DW) [8]. Borja-Martínez et al. [31] studied the α-tocopherol content in different mixtures of by-products (leaves and stems) from two varieties (Parthenon and Naxos), highlighting the combination of both as having the highest content (9.45 mg/g).

Vitamin K is also significant in broccoli, particularly in the leaves compared to the stems, due to its phylloquinone content [8]. Accurate quantification of vitamin K forms in broccoli (stems) has been achieved through optimized LC-ESI-MS/MS methods. These methods allow for the detection and measurement of phylloquinone and various menaquinones with high precision and low limits of quantification [52].

### 3.6. Minerals

The mineral content in broccoli varies depending on the part of the plant analyzed. Stems exhibit higher levels of minerals such as magnesium and sodium compared to leaves [8]. Conversely, minerals like calcium and iron are found in greater quantities in the leaves of the “Green Magic” broccoli variety [8]. AL-Altaie and Addai [35] extracted minerals from broccoli stems using a HNO_3_:HCIO_4_ (2:1) solution for 3–4 h at 100 °C, highlighting significant magnesium (364.67 mg/100 g DW) and calcium (1245.34 mg/100 g DW) content, as analyzed by atomic absorption spectrometry (AAS). These findings are consistent with those of Núñez-Gómez et al. [34], who reported high levels of magnesium (2555.2 mg/kg DW) and calcium (4887.8 mg/kg DW) in the stems, along with substantial sodium content (2910.1 mg/kg DW). Additionally, other minerals such as zinc, copper, phosphorus, and potassium are notable in broccoli by-products [8,34].

Table 3 summarizes the data on pigments, vitamins, and minerals from the reviewed articles on broccoli by-products. The data are organized according to the processing methods applied prior to plant analysis, as well as the extraction and determination techniques used.

### 3.7. Proteins

Broccoli by-products exhibit significant protein content, which varies depending on the plant part. Broccoli leaves contain higher protein levels, with 22.75 g/100 g DW when lyophilized and 6.13% when fresh [21,53]. Lyophilized stems have a protein content of 14.37–14.10 g/100 g [33,38], while fresh sprouts contain 0.86 g/30 g [21]. Additionally, Shi et al. [54] reported higher protein content in juice and puree from combined broccoli by-products (stems and leaves) compared to the pulp.

The primary proteins in broccoli are albumin and globulin, which are abundant in the leaves (31.66% and 15.89%, respectively), along with other proteins such as glutelin and prolamin [53].

#### Amino Acids

Amino acids, the building blocks of proteins, play a crucial role in cellular biochemistry and various metabolic reactions. A wide variety of free amino acids have been identified in broccoli by-products, particularly in sprouts. Some of these amino acids include lysine, l-histidine, d-serine, l-asparagine, l-leucine, l-tryptophan, and glycitin, among others [15,26]. The most common extraction method for these compounds is solvent extraction (ethanol), followed by analysis using UHPLC-QTOFMS/MS or GC-EIMS.

### 3.8. Lipids

Lipids are a minor component in broccoli by-products. Fatty acids are the basic components of some lipids, so their content in broccoli by-products has been extensively studied. Rivas et al. [12] extracted higher amounts of total fatty acids from the leaves of the “Parthenon” variety (3.34 g/100 g DW), slightly more than from the stems (3.17 g/100 g DW) and the inflorescence (2.48 g/100 g DW). However, the fatty acids extracted from fresh sprouts were even lower (0.13 g/30 g) [21]. Additionally, Shi et al. [54] analyzed the total lipid content in puree, pulp, and juice from broccoli stems and leaves, finding no significant differences (7.3–8.1%).

The processing method prior to extraction affects the concentration of fatty acids. Villaño et al. [38] demonstrated that air-drying stems at 60 °C resulted in less degradation of fatty acids compared to air-drying at 80 °C or lyophilization.

The primary fatty acids in broccoli by-products are palmitic acid and linolenic acid, which are more abundant in the inflorescences. Other identified fatty acids include 8,15-DiHETE, FA18:2 + 3O, FA 18:4 + 2O, and 9-HODE, among others [11,33].

### 3.9. Carbohydrates

Total carbohydrates in broccoli consist primarily of oligosaccharides and polysaccharides, which are part of dietary fiber, as well as simple sugars, with small amounts of starch. Total carbohydrates in broccoli are often determined by differences from other components, such as moisture, protein, and ash [38,54]. Total carbohydrate values vary significantly depending on the part of the vegetable studied and the processing method, with a maximum of 82.27 g/100 g DW in lyophilized stems [34] higher than in broccoli sprouts (0.09 g/30 g FW) [21].

Sugars, although present in broccoli carbohydrates, are found in lower amounts compared to other vegetables. Among broccoli by-products, leaves (25.39 g/100 g DW) and inflorescences (24.61 g/100 g DW) have the highest free sugar content (glucose, fructose, and sucrose), followed by stems (18.58 g/100 g DW) [12].

Dietary fiber, abundant in broccoli, is not fully digested by human enzymes and is classified as total dietary fiber (TDF), insoluble dietary fiber (IDF), or soluble dietary fiber (SDF). TDF primarily includes uronic acid and neutral sugars such as glucose, rhamnose, and galactose [12,34]. IDF is primarily composed of cellulose, hemicellulose, and lignin, and SDF includes pectins, gums, mucilages, and beta-glucans.

Total fiber determination is conducted using the enzymatic–gravimetric method, employing enzymes such as protease, amyloglucosidase, and α-amylase [34,40,54]. Using this method, the total fiber content in broccoli stems has been determined to be 72.28 g/100 g DW, higher than in leaves (62.22 g/100 g DW) and inflorescences (64.42 g/100 g DW) [12]. Villaño et al. [21] reported a total fiber content of 0.74 g/30 g FW in broccoli sprouts. The processing method did not significantly alter the total fiber content in broccoli stems when subjected to hot air-drying at 60 and 80 °C (22.87, 22.34 g/100 g DW); however, the fiber content extracted from lyophilized stems was lower (17.71 g/100 g DW) [21]. As with total fiber content, broccoli stems contain higher amounts of IDF (56.27–66.18 g/100 g DW) and SDF (5.94–11.10 g/100 g DW) compared to leaves and inflorescences, with IDF being much more prevalent [12]. However, the IDF and SDF content determined by Schäfer et al. [55] and Núñez-Gómez et al. [34] in their studies was lower, due to their determination in lyophilized and fresh stems, respectively, unlike in the work of Rivas et al. [12], who dried the samples with air at 45 °C. The IDF and SDF content is also influenced by the broccoli variety, being lower in the “Beneforte” variety compared to “Parthenon”, with differences of up to double for both components [12,40].

Uronic acid plays a crucial role in the structure of plant cell walls and significantly contributes to the total sugar content (30.1%) [34], especially in the form of pectin. Galactose, although in lower proportion compared to uronic acid, is an important component found in both SDF (2.5–3.1% in leaves) and IDF (10.7% in stems and 1.2–1.5% in leaves), with a notable presence in TDF (11.4% in stems) [34,40]. However, the concentration of glucose in the vegetable (broccoli stems) is much higher than that provided solely by dietary fiber, at 10% [34]. Rhamnose and glucose, although in lower concentrations than galactose and uronic acid, are also part of the polysaccharide matrix of broccoli cell walls and contribute to the total sugars present. The rhamnose and glucose content in TDF from broccoli stems is 1.7% and 3.6%, respectively, with 2.3% and 4.0% corresponding to IDF [34]. In broccoli leaves, the glucose percentage is higher in IDF (1.0–12.2%) than in SDF (0.7–1.1%), while the rhamnose percentage is higher in SDF (4.1–5.9%) [40]. Petkowicz and William [1] also determined the content of galactose, rhamnose, and galacturonic acid in the pectin fraction of broccoli stems, obtaining values of 13.6%, 5.4%, and 74.7%, respectively.

The analytical methods used to determine sugars include colorimetric techniques, gas chromatography (GLC-FIC), high-performance liquid chromatography (HPAEC-PAD), and UV–Vis spectroscopy. These methods allow the identification and quantification of a wide variety of sugars in addition to those already mentioned, such as fucose, arabinose, xylose, and mannose [1,34,40].

### 3.10. Others Compounds

Hydrocarbons, although present in minimal amounts, are another class of compounds identified in broccoli. These compounds contribute to the phytochemical profile of broccoli and can influence its antioxidant properties and health benefits. Abdelhalim et al. [50] identified several hydrocarbons in fresh broccoli leaves using GC-MS. The content of these compounds varied depending on the extraction method employed. Extracts rich in 3-methyloctacosane and 15-methyltriacontane were obtained using hexane as a solvent, whereas maceration with dichloromethane (DCM) for 7 days yielded other hydrocarbons such as 3,3,17,17-tetraethylnonadecane and 13-methylnonacosane.

These components not only determine the nutritional value of broccoli but also its functional properties in the food industry and nutraceutical applications. Investigating the variation in the chemical composition of broccoli by-products under different processing conditions, as well as utilizing a wide range of analytical methods, is crucial for their potential use in the food industry and nutraceutical applications.

All data related to proteins, lipids, carbohydrates, and other compounds present in broccoli by-products, as well as extraction and analysis methods, are listed in Table 4.

## 4. Beneficial Properties

Different broccoli by-products and the bioactive compounds they contain have been described as having various beneficial effects, such as antioxidant, antimicrobial, anticancer, anti-inflammatory and antihypertensive activities, as well as effects on gut health (Table 2).

### 4.1. Antioxidant Activity

Broccoli by-products, including leaves and stems, exhibit significant antioxidant properties that can be harnessed to enhance health. Various parts of the broccoli plant are rich sources of bioactive compounds. Ethanolic extracts of broccoli by-products have demonstrated high antioxidant capacity, attributed to their substantial polyphenol content [26,56]. Notably, broccoli leaves have shown superior antioxidant capacity, primarily assessed using the DPPH and ABTS methods [11,32].

Methanol is the most effective solvent for extracting broccoli compounds with high antioxidant activity, outperforming water extraction [57]. Metabolomic analysis has identified unique phytochemicals in methanolic extracts of leaves and stems, with leaves being particularly rich in flavonoids and coumarins, which contribute to their superior antioxidant activity [33]. Methanol, combined with formic acid, has also been used to extract compounds from broccoli stems, yielding high antioxidant values as measured by the FRAC and ORAC methods [34]. Other solvents, such as acetone [35] and supercritical fluid extraction [31], have also been employed, with supercritical fluid extraction showing greater antioxidant potential. Hexane maceration of by-products (leaves and stalks) yields extracts rich in volatile compounds with antioxidant properties [58].

Martínez et al. [36] utilized water as a solvent to obtain antioxidant extracts from broccoli by-products, investigating the effects of ultrasound and enzymatic treatments during extraction. The antioxidant capacity of these extracts was analyzed using in vivo electrical impedance under oxidative stress conditions. Enzymatic extraction technology produced extracts with higher total polyphenol content, leading to a greater reduction in the growth of the *Saccharomyces* strain.

The pre-treatment of broccoli also affects the extracted compounds and, consequently, the antioxidant activity of the extracts. Villaño et al. [38] found that drying at 80 °C was more effective than freeze-drying in obtaining higher antioxidant activity in the stems.

Gudiño et al. [11] examined the variation in antioxidant activity of by-products based on the broccoli variety, with Summer Purple showing the highest activity among the five varieties analyzed. Borja-Martínez et al. [31] compared the antioxidant activity of extracts from leaves and stems of the Parthenon and Naxos varieties obtained via supercritical fluid extraction. The results indicated that the leaves of the Naxos cultivar exhibited higher antioxidant activity, highlighting significant variability in antioxidant activity depending on the broccoli cultivar.

Broccoli by-products are also rich in proteins and fiber, with high antioxidant capacity, making them suitable for use as food ingredients [34,54]. The juice fraction of broccoli leaves and stems has shown higher polyphenolic content and antioxidant activity compared to the pomace fraction [54]. The antioxidant activity of these extracts is also attributed to the presence of sulforaphane, particularly in broccoli seeds and sprouts [9].

### 4.2. Antimicrobial Activity

The antimicrobial properties of broccoli by-products are significant. Extracts from these by-products (stems, leaves, sprouts, etc.) have demonstrated efficacy against various pathogens, including *Listeria*, *Salmonella*, *Bacillus cereus*, and *Staphylococcus aureus* [11,32,36,42]. This antimicrobial activity is attributed to bioactive compounds such as phenolics, glucosinolates, and fatty acid derivatives [11,32]. Martínez et al. [36] linked the high inhibitory capacity of broccoli extracts (leaves and stems) against *Listeria* sp. to the presence of chlorogenic acid, obtained through enzymatic extraction (EE). Additionally, peptides from broccoli seeds have shown activity by damaging the cell membranes of fungi like *Colletotrichum gloeosporioides* and reducing hyphal growth. These peptides also exhibited antibacterial activity by inhibiting pathogenic bacteria without affecting probiotic bacteria such as *Lactobacillus casei* [59]. Peptides in stem extracts inhibited the growth of bacteria like *Staphylococcus xylosus* by up to 60% [60].

The antimicrobial capacity of leaves has been extensively studied. Their activity varies depending on the solvent used for extraction. Hexane or DCM yields extracts effective against *Escherichia coli* and *S. aureus*, with hexane-extracted compounds showing greater effects [50]. However, the activity is lower when aqueous solvents are used [58]. Cao et al. [61] employed different extraction methods on leaves, combining ethanol and a solvent composed of choline chloride and propylene glycol (ChCl-PG) with ultrasound (UAE). The UAE + ChCl-PG method produced extracts with the highest phenolic content, rich in hydroxycinnamic acids, responsible for the inhibitory effect against *S. aureus*, *E. coli*, and *Salmonella* sp.

For stems, acetone extraction proved effective against *Klebsiella* sp., *E. coli*, *S. aureus*, and *Pseudomonas aeruginosa* [35]. Despite the effectiveness of leaves, stems can exhibit greater activity against *S. aureus* when aqueous solvents are used [58]. Gudiño et al. [11] achieved higher inhibition percentages against *B. cereus*, *S. aureus*, and *Listeria* sp. using extracts from stems and inflorescences compared to leaf extracts.

Combined extracts of broccoli leaves and seeds also demonstrated activity against *Bacillus subtilis* and *Salmonella typhimurium* (0.39–0.78 mg/mL). However, the combination of by-products showed low activity against *S. aureus* and *E. coli* [32].

### 4.3. Anticancer Activity

The non-edible parts of broccoli contain bioactive compounds that exhibit significant anticancer activity. Broccoli seed flours have been shown to inhibit the proliferation of prostate cancer cells (LNCaP) by up to 20% after 96 h of treatment. This anticancer effect is attributed to specific compounds inherent to broccoli, such as the glucosinolate glucoraphanin, a precursor of the isothiocyanate sulforaphane, which possesses anticancer properties [62].

The cancer-preventive properties of cruciferous vegetables depend on the specific cultivars and the ability to extract their bioactive glucosinolates, which is influenced by storage, cooking, and other post-harvest processes, as well as cultivation practices. The application of CO_2_ during cultivation increases the glucoraphanin content in broccoli sprouts and enhances the activity of the enzyme myrosinase, which hydrolyzes glucosinolate to sulforaphane. This CO_2_-induced enhancement of compounds in the sprouts improves the vegetable’s anticancer properties.

While the cytotoxic activity of broccoli by-products is primarily associated with glucosinolates and isothiocyanates, phenolic compounds also contribute to this effect. Kim et al. [63] demonstrated that chloroform and hexane fractions of sprout extracts reduced the viability of breast cancer stem cells, attributing this to their high content of phenolic compounds and oleic acid. In another study, Nicolas-Espinosa et al. [64] extracted bioactive peptides from the cell membrane (MF) and total proteins (E) of broccoli stems. These peptides, when applied to human keratinocytes, allowed controlled proliferation, improved the wound healing process in the skin, and prevented the development of cancer cells. Broccoli sprouts and seeds exhibited high cytotoxic effects against cancer cell lines (lung, colon, and liver) at low concentrations. Additionally, these extracts induced apoptosis in lung cancer cells (Caco-2) [32,65]. The activity of the seeds was slightly lower than that of the sprouts, due to a lower content of phenolic compounds (gallic acid, esculetin, quercetin) and vitamin C. Methanolic extracts from the non-edible parts of broccoli (stems and leaves) showed cytotoxic effects on HePG2 (42.82%), MCF7 (60.43%), and HCT116 (31.28%) when applied in vitro at 100 μg/mL. This effect is attributed to the presence of phenolic compounds such as flavonoids and tannins, as well as vitamin C, and was greater than that observed in other *Brassica* family vegetables like cauliflower or radish [57].

### 4.4. Anti-Inflammatory Activity

Broccoli by-products, including stems, leaves, and inflorescences, have been identified as rich sources of valuable bioactive compounds. The aqueous extract of broccoli sprouts, which is high in glucoraphanin, has been shown to inhibit the formation of harmful advanced glycation end-products (AGEs), thereby reducing inflammation and oxidative stress in endothelial cells and improving vascular function [66].

Park et al. [67] investigated the effects of broccoli leaf extracts on immune responses related to inflammation. Their findings indicated that the extracts inhibited cytokine activity and the NF-kB signaling pathway, which is responsible for initiating and resolving inflammatory responses. In another study, Choe et al. [62] demonstrated that seed flour extracts exhibited potential anti-inflammatory properties, inhibiting IL-1β mRNA expression by 38.9% compared to the control.

Ramesh et al. [58] compared the anti-inflammatory efficacy of broccoli stem and leaf extracts with aspirin. They found that stem extracts provided significant membrane stabilization at concentrations of 500 μg/mL, preventing cell lysis and achieving high proteinase inhibition (78%). These results underscore the superior anti-inflammatory capacity of broccoli stem extracts compared to leaf extracts and aspirin.

As with its anticancer properties, the application of CO_2_ during broccoli cultivation was found to increase sulforaphane content in sprouts, thereby enhancing their anti-inflammatory effects by inhibiting cyclooxygenase (COX) and lipoxygenase (LOX) activities [68].

In vivo studies have also been conducted in both animals and humans. Khalil et al. [69] administered aqueous extracts of broccoli leaves to rats to mitigate the side effects of gentamicin, such as nephrotoxicity and hepatotoxicity. The extracts demonstrated anti-inflammatory properties by reducing the expression of key inflammatory genes and inflammation in the affected renal and hepatic tissues. Similarly, Sim et al. [70] explored the anti-inflammatory effects of broccoli sprout extract in a mouse model of lipopolysaccharide-induced liver damage. The extract provided a protective effect against inflammation and liver damage, increasing the survival rate of treated mice. Additionally, López-Chillón et al. [71] reported that consuming broccoli sprouts for 10 weeks reduced inflammation levels and C-reactive protein in overweight individuals.

### 4.5. Antihypertensive Activity

Recent research has highlighted the potential of broccoli-derived peptides in managing hypertension and associated cardiovascular conditions. Several hydrolyzed peptides isolated from proteins in broccoli stems and leaves have demonstrated significant in vitro angiotensin-converting enzyme (ACE)-inhibitory activity [72,73]. Additionally, other bioactive compounds found in broccoli by-products have been linked to antihypertensive effects. Gudiño et al. [11] observed ACE-inhibitory activity exclusively in leaf extracts, attributing this effect to phenolic compounds unique to that part of the plant. Sulforaphane, has also been shown to enhance endothelial function and lower blood pressure in pregnant women with hypertension [74].

These findings underscore the promising role of bioactive compounds from broccoli by-products in the development of natural antihypertensive agents and functional foods aimed at promoting cardiovascular health.

### 4.6. Effects on Gut Health

Broccoli consumption has been shown to have beneficial effects on gut health and metabolism. Studies indicate that broccoli extract can increase probiotic bacteria and inhibit harmful bacteria in the gut [44]. Regular consumption of broccoli sprouts has been shown to promote the growth of beneficial intestinal bacteria, such as Lachnospiraceae and Ruminococcaceae, helping to reduce inflammation and strengthen the intestinal barrier in mouse models, an effect attributed to sulforaphane [20]. Similarly, dietary fibers derived from broccoli stems have been shown to enhance the growth of beneficial bifidobacteria while reducing lactic acid bacteria in the colon, thanks to their prebiotic properties and ability to produce short-chain fatty acids [75]. Additionally, Costa-Perez et al. [76] determined that the SFNs present in broccoli stems possess outstanding anti-inflammatory potential, associated with diseases such as ulcerative colitis and Crohn’s disease. This is manifested in their ability to significantly reduce the levels of proinflammatory cytokines (IL-6, IL-8, and TNF-α) in an in vitro model of intestinal inflammation. These studies suggest that broccoli consumption may have potential health benefits through its effects on gut microbiota and metabolic markers, although further research is needed to fully understand the mechanisms involved.

Beneficial effects of different broccoli by-products are shown in Table 5.

## 5. Brassica By-Products Applications

Due to their richness in bioactive compounds and the numerous benefits they confer, broccoli by-products have diverse applications across various fields, including food, cosmetics, and medicine (Figure 1).

### 5.1. Food Formulation

Given the high concentration of bioactive compounds in broccoli, numerous studies have explored its incorporation into food products to enhance their quality and nutritional value. Drabińska et al. [77] incorporated broccoli leaf powder into wheat pasta, finding that a 5% addition enriched the pasta with volatile compounds and minerals without significantly affecting its technological or sensory qualities. Similarly, Devi et al. [78] investigated the use of broccoli leaves as an additional ingredient in noodle production to increase fiber and protein content. The addition of 5% broccoli leaves resulted in noodles with a fiber content of up to 0.62% and a protein content of up to 14.14%, while also reducing fat and carbohydrate content compared to the control. Kamiloglu et al. [79] incorporated broccoli leaf and stem powder into salad dressings based on olive, hazelnut, sunflower oils, and lemon juice, finding that the oily matrix facilitated the high bioavailability of broccoli’s phenolic compounds. Lucera et al. [80] examined the physicochemical and sensory properties of spreadable cheese enriched with broccoli stem and leaf flour, among other vegetable by-products.

A significant portion of broccoli’s bioactive compounds are responsible for its antioxidant and antimicrobial activities. Incorporating these compounds as natural additives into other foods can extend the shelf life of the final product. Castillejo et al. [81] studied the effect of adding broccoli leaves and mustard as supplements in kale pesto sauce, finding that the addition increased phenolic compound content and antioxidant activity, which, along with mustard’s antimicrobial properties, extended the product’s storage life to up to 20 days at 5 °C. Similarly, Saavedra-Leos et al. [82] proposed encapsulating juice from broccoli leaves and stems using 5% maltodextrin as an effective method to incorporate broccoli’s bioactive compounds into other foods, aiming to enhance the phenolic content and antioxidant capacity of the new food matrix, thereby extending its shelf life.

Ferreira et al. [83] suggested incorporating extracts from broccoli by-products (stems, leaves, and inflorescences) into béchamel sauce. They used microwave hydrodiffusion and gravity technology to dehydrate broccoli by-products and recover water-soluble compounds, which were then incorporated into the sauce instead of water, providing bioactive compounds such as glucosinolates.

The diagnosis of celiac disease has increased, largely due to improved diagnostic techniques and greater societal awareness. Consequently, the consumption of gluten-free foods has risen, not only among individuals with celiac disease but also among those without the condition. However, many gluten-free foods have lower nutritional value, often lacking essential amino acids. Drabińska [84] investigated the incorporation of broccoli leaf powder, rich in amino acids, as a solution to this deficiency. In this study, broccoli leaf powder was used in gluten-free cakes, replacing corn and potato starch in varying proportions. The addition of 2.5% broccoli leaf powder as a starch substitute increased the protein and mineral content of the cakes, as well as the content of phenolic compounds and glucosinolates, thereby enhancing antioxidant capacity [85]. Krupa-Kozak et al. [86] and Krupa-Kozak et al. [28] incorporated broccoli leaves into gluten-free bread, increasing protein and mineral content and improving the bread’s antioxidant potential. The inclusion of broccoli by-products in bread preparation also improved its appearance after baking by increasing its volume and preventing baking losses. However, Lafarga et al. [87] found that adding 2% (*w*/*w*) broccoli stem and leaf powder to bread formulations decreased the weight and volume compared to control bread. Nonetheless, the incorporation of broccoli by-products significantly increased the total phenolic content and antioxidant capacity of the breads.

### 5.2. Substitute Additive

Extracts from broccoli by-products can be utilized in the food industry as natural alternatives to conventional additives, owing to the presence of compounds that perform similar functions. A study by Petkowicz and William [1] investigated the high thickening and emulsifying capacities of pectins extracted from broccoli stems, proposing their use as alternatives to other pectins in food applications. Mohammed et al. [88] replaced synthetic antioxidants with natural ones in the production of lentil soup powder by incorporating nanoencapsulated extracts from broccoli inflorescences. These nanoencapsulations demonstrated high thermal stability and significant antioxidant and anticancer activities, attributed to the richness of the extracts in phenols, flavonoids, and glucosinolates. Angiolillo et al. [30] incorporated extracts from broccoli stems and leaves (SFEs) as an eco-friendly alternative to enhance the preservation of fresh filled pasta. The extracts extended the shelf life of the pasta by 6 days compared to the control, enriching it with phenolic compounds.

Nitrate and nitrite salts are also important food additives, particularly for improving the quality and shelf life of meat products. Broccoli stems and inflorescences are significant sources of nitrates, which can be added to meat products as natural substitutes for E-type additives [89,90].

### 5.3. Snacks

Ying et al. [91] developed snacks using freeze-dried broccoli leaf powder combined with pomace (3%). These ingredients were mixed with rice flour, CaCO_3_ (1%), and NaCl (0.5%). The resulting dough underwent an extrusion process with a moisture content of 20%, followed by heating at 60 °C for 4 h (water activity < 0.2 aw). This process produced a final product with high protein and dietary fiber content, rich in phenolic compounds.

Fanesi et al. [92] utilized florets and stems to produce broccoli flour, which was then used to make cookies enriched with phenolic compounds (1.9 mg GAE/g DW), glucosinolates (33 μg/g DW), and carotenoids (46 μg/g DW).

### 5.4. Beverages

Salas-Millán et al. [93] employed the fermentation of broccoli leaves to create a novel beverage fermented by *Lantiplantibacillus plantarum* through fermentation-assisted extraction. The pasteurization process increased the polyphenol content in the beverage but reduced the bacterial population (3.5 CFU/mL). However, the application of high hydrostatic pressure resulted in a higher sulforaphane content and maintained bacterial population stability throughout the beverage’s shelf life. Similarly, Pérez et al. [94] developed a vegetable drink from broccoli stems and carrots, utilizing two processing treatments: pasteurization and ultrasound. The ultrasound treatment preserved the antioxidant capacity of the beverage during cold storage, maintaining the levels of phenolic compounds and carotenoids, and increasing the sulforaphane nitrile content by 14% compared to the untreated control.

Sánchez-Bravo et al. [95] developed craft beers supplemented with broccoli sprouts and by-product powders. These beers exhibited high concentrations of the isothiocyanate sulforaphane, with 5.00 mg/L in those supplemented with sprouts and 2.54 mg/L in those with by-product powder. After bottling, these concentrations decreased to 1.08 and 0.03 mg/L, respectively, after 150 days of storage.

### 5.5. Encapsulation of Active Compounds

Encapsulation is a widely utilized technique in food processing, enhancing the stability and bioavailability of bioactive compounds. Encapsulated products offer significant health benefits, making them ideal functional ingredients for the food industry. The primary methods for encapsulating compounds include spray-drying and freeze-drying, using materials such as maltodextrin, gum arabic, cellulose, or soy protein as wall materials [96].

Extracts from broccoli by-products were encapsulated with 10% maltodextrin as the wall material, using a core-to-wall material ratio of 1:2 at 80 °C. These encapsulates were added at 5% (*w*/*w*) to fish burgers, enriching them with higher total phenol and flavonoid content, as well as greater antioxidant activity compared to the control. It was also determined that the properties of the added extracts were not significantly affected by the encapsulation process or by cooking [97].

### 5.6. Other Applications

In the food industry, extracts from broccoli seeds (0.7%) have been incorporated into the production of biodegradable films, combined with carboxymethylcellulose and polyvinyl alcohol. This film was applied to butter packaging, improving its ability to preserve acidity and reducing color loss during cold storage [98].

Beyond their use as a source of active compounds in the food industry, broccoli by-products can be applied in other fields such as environmental science and cosmetics. Granado-Castro et al. [99] used broccoli stems as biomass to remove toxic metals like Pb(II) from aqueous solutions. The stems, rich in hemicellulose, starch, pectin, cellulose, lignin, and glucosinolates, enable Pb(II) sorption by acting as electron donors. The biomass generated from the stems showed a maximum adsorption capacity of 586.7 mg/g. The application of this biomass in sorption processes offers an eco-friendly solution for cleaning contaminated environments, applicable in wastewater treatment or industrial effluents. Another potential application is the incorporation of extracts from leaves and stems as ingredients in cosmetic formulations. Their high antioxidant activity and ability to mitigate the negative effects of UV light on cells make them beneficial for skin products [31].

## 6. Conclusions

Broccoli by-products, often considered the non-edible parts of the vegetable, offer a comprehensive nutritional and functional profile. Glucosinolates, particularly glucoraphanin, are characteristic compounds found in notable concentrations in broccoli stems and sprouts. Sulforaphane, a degradation product of glucoraphanin, is the most abundant isothiocyanate in broccoli, recognized for its potent antioxidant and anticancer properties. Broccoli leaves are rich in phenolic compounds (kaempferol and quercetin), pigments (chlorophyll and carotenoids), vitamins (C, E, and K), and essential minerals (Mg, Ca). Additionally, broccoli by-products provide proteins, amino acids, lipids, and carbohydrates. The leaves contain the highest protein content, including essential amino acids such as lysine and leucine, and bioactive peptides. Lipids are present in smaller quantities, primarily as unsaturated fatty acids like linolenic acid. Stems are an excellent source of dietary fiber and sugars such as galactose and glucose, which are important for digestive health and metabolic control.

The concentration, extraction, and effectiveness of bioactive compounds from broccoli by-products are significantly influenced by processing and extraction techniques. Methods such as ultrasound-assisted extraction and freeze-drying are noted for their efficiency and sustainability. Studies have shown that these techniques significantly impact the quantity and quality of the obtained bioactive compounds. Ultrasound-assisted extraction and the use of solvents like methanol and ethanol are particularly effective in extracting phenolic compounds and glucosinolates. Advanced analytical methods such as high-performance liquid chromatography (HPLC) and mass spectrometry (MS) have enabled precise characterization of these bioactives.

Due to their richness in bioactive compounds, broccoli by-products exhibit significant beneficial properties. Leaf extracts obtained using solvents like methanol and ethanol are notable for their substantial antioxidant capacity. In terms of antimicrobial activity, stem and leaf extracts have shown effectiveness against pathogens such as *Listeria*, *Salmonella*, and *Staphylococcus aureus*, attributed to their high phenolic content. Sulforaphane provides high anticancer capacity by inhibiting the proliferation of cancer cells and promoting apoptosis in various types of cancer (lung, colon, prostate). Various studies also support the anti-inflammatory capacity of different parts of broccoli, reducing inflammation, improving vascular health, and preventing liver and kidney damage.

Broccoli by-products, such as leaves and stems, have become valuable sources of bioactive compounds with diverse applications in the food industry. Their richness in antioxidants, fibers, proteins, minerals, and glucosinolates allows them to enhance the nutritional quality of food products, extend shelf life, and serve as functional ingredients. They have also been used as natural substitutes for additives, thickeners, and preservatives, promoting healthier and more sustainable food consumption. This underscores the functional value of broccoli parts that are often underutilized or discarded. This literature review reveals that utilizing these by-products promotes sustainability by reducing agricultural waste and fostering a circular economy.

In future research, it would be relevant to delve deeper into the possible therapeutic and preventive applications of the bioactive substances in broccoli, as well as the analysis of their interactions with other dietary components and the stability of these compounds during processing, to guide their practical application in the field of health.

## Figures and Tables

**Figure 1 foods-13-03918-f001:**
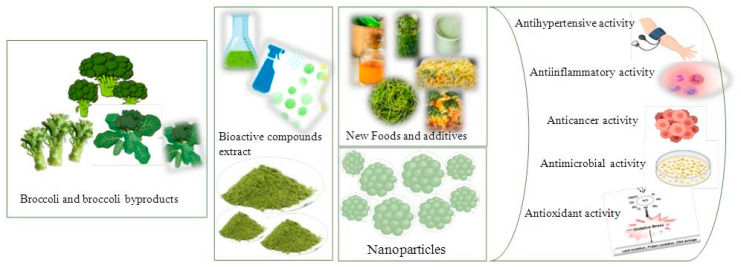
Visual overview of future perspectives on use of broccoli and broccoli by-product extracts.

**Table 1 foods-13-03918-t001:** Phenolic compounds and organic acids in diverse broccoli by-products.

Compound	Units	By-Products Content	Vegetable Processing	Extraction Method	Analysis Method	Ref.
**Phenolic Compounds**						
**Total phenolic content (TPC)**	mg GAE/100 g DW	2435 (l); 939 (st); 1074 (if)	forced air oven (45 °C, 48 h)	UAE: EtOH (80% *v*/*v*); 45 °C, 1 h	UV–Vis	[11]
	mg GAE/100 g DW	385.4 (sp); 480.4 (s)	dried oven	EtOH (70%, 1:20 *w*/*v*) orbital shaker, 50 °C, 1 h	UV–Vis	[26]
	mg GAE/100 g DW	79.22 (st)	dried (60 °C, 24 h)	Acetone (50%)	UV–Vis	[35]
	mg GAE/g	74.31 (l + st)	dried in a climatic chamber (55 °C, 24 h)	MeOH:Tris–HCl (50 mM, pH 7.5) (1:1 *v*/*v*), 4 °C, 20 min	UV–Vis	[31]
		19.51 (l + st) (P)		SFE: CO_2_ 40 °C, 443 bar, 7% eToh, 31 g/min of flow rate (68 min)	UV–Vis	
		29.21 (l + st) (N)			UV–Vis	
	mg GAE/g DW	28.50 (l); 16.55 (s)	dried oven (40 °C)	MeOH (70%)	UV–Vis	[32]
		25.77 (l); 15.096 (s)		EtOH (70%)	UV–Vis	
		24.79 (l); 12.58 (s)		Hot water	UV–Vis	
	mg GAE/g DW	1.8 (st)	osmotic dehydration	MeOH (80%), 2 h	UV–Vis	[37]
		0.6 (st) (40 °C), 0.45 (st) (60 °C)	MW–hot air (40–60 °C)		UV–Vis	
	μg/g DW	10,605.5 (l + st)	freeze-drying	MeOH (70%) 70 °C, 30 min	UHPLC-DAD-ESI/MS	[39]
		7419.7 (l + st)	air-drying (40 °C)			
		15,086.7 (l + st)	blanching and freeze-drying			
		4588.7 (l + st)	blanching and air-drying (40 °C)			
		2203.0 (l + st + if)	freeze-drying			
		2765.5 (l + st + if)	MW hydrodiffusion and gravity			
	mg GAE/100 g Extract	0.70 (l); 0.17 (st); 0.79 (if)	forced air oven (45 °C, 48 h)	UAE: EtOH (80% *v*/*v*); 45 °C, 1 h	UV–Vis	[12]
	mg GAE/g DW	4.14 (l); 1.41 (st)	freeze–drying	MeOH (70%), 75 °C	UV–Vis	[8]
	mg GAE/g DW	3.8 (st); 6.1 (rf)	freeze–drying	UAE: EtOH (80% *v*/*v*); 37 °C, 20 min	UV–Vis	[10]
		5.4 (st + rf)				
	mg GAE/g FW	0.16–2.44 (sp) (3–7 days); 3.89–4.58 (s)	blanched (55 °C, 15 min)	MeOH, 30 min	UV–Vis	[9]
	mg/g	»9 (l); »5 (st)	freeze–drying	MeOH (80%), HCl (1%); 2 h	UV–Vis	[33]
	mg GAE/100 g DW	154.7 (st) (P)	freeze–drying	MeOH–Water–Formic acid (79/19/1 *v*/*v*/*v*)	UV–Vis	[34]
	mg/kg DW	21.32 (l + st)	Fresh	Water (1:3 *w*/*w*); 100 °C; 60 min	UV–Vis	[36]
		20.95 (l + st)		EE: water (1:3 *w*/*w*), cellulose (0.01%)	UV–Vis	
		3.91 (l + st)		UAE: Water (1:3 *w*/*w*); 75 °C; 60 min	UV–Vis	
	mg GAE/100 g DW	112.95 (l)	dried (50 °C, 24 h)	DM: EtOH (30%); 50 °C; 36, 92 min	UV–Vis	[15]
	mg GAE/g DW	3.01 (st)	freeze–drying	EtOH (70%; 25 °C; 30 min)	UV–Vis	[38]
		3.78 (st)	air–drying (60 °C)		UV–Vis	
		8.37 (st)	air–drying (80 °C)		UV–Vis	
* Flavonoid *						
**Total Flavonoid Content (TFC)**	mg CE/100 g	206.9 (sp); 216.9 (s)	dried oven	EtOH (70%, 1:20 *w*/*v*) orbital shaker, 50 °C, 1 h	UV–Vis	[26]
	mg CE/g DW	8.71 (l); 3.74 (s)	dried oven (40 °C)	MeOH (70%)	UV–Vis	[32]
		9.93 (l); 3.51 (s)		EtOH (70%)	UV–Vis	
		7.84 (l); 3.59 (s)		Hot water	UV–Vis	
	mg QE/g DW	2.4 (st); 5.4 (rf)	freeze–drying	UAE: EtOH (80% *v*/*v*); 37 °C, 20 min	UV–Vis	[10]
		4.3 (st + rf)			UV–Vis	
	mg RE/g FW	0.46–3.02 (sp) (day 3–7); 2.60–4.01 (s)	blanched (55 °C, 15 min)	MeOH, 30 min	UV–Vis	[9]
	mg/g	3.4 (l); 0.9 (st)	freeze–drying	MeOH (80%), HCl (1%); 2 h	UV–Vis	[33]
**K-3-o-sophoroside**	mg/kg DW	ND (st); 39.4 (rf)	freeze-drying	UAE: MeOH (100% *v*/*v*); 75 °C, 15 min	UPLC-MS/MS	[10]
**Q-3-diglucoside-7-glucoside**	mg/kg DW	ND (st); 17.2 (rf)	freeze-drying	UAE: MeOH (100% *v*/*v*); 75 °C, 15 min	UPLC-MS/MS	[10]
**Quercetin**	mg/g Extract	0.97 (l); 0.56 (s)	dried oven (40 °C)	EtOH (70%)	HPLC	[32]
**Myricetin**	mg/g Extract	2.77 (l); 0.31 (s)	dried oven (40 °C)	EtOH (70%)	HPLC	[32]
* Coumarins *						
**Esculetin**	mg/g Extract	6.49 (l); 10.18 (s)	dried oven (40 °C)	EtOH (70%)	HPLC	[32]
* Phenolic acids *						
**Total Phenolic Acids**	mg/g DW	3.6–5.7 (l) (B)	air-dried	EUA: MeOH (60%), 35 °C, 1 h; alkaline hydrolisis	HPLC-DAD-ESI-MS	[40]
* **Benzoic acids** *						
**Vanillic acid**	μg/mL	ND (l); 17.82 (st)	freeze-drying	UAE: EtOH (50%)	HPLC	[41]
**Gallic acid**	mg/g Extract	0.15 (l); 0.17 (s)	dried oven (40 °C)	EtOH (70%)	HPLC	[32]
	μg/mL	ND (l); 17.13 (st)	freeze-drying	UAE: EtOH (50%)	HPLC	[41]
* **Hydroxycinnamic acids** *						
**Caffeic acid**	mg/g DW	213.2 (sp)	freeze-dried	MeOH (70%, *v*/*v*); 70 °C, 30 min	HPLC	[42]
	mg/g Extract	ND (l); 0.12 (s)	dried oven (40 °C)	EtOH (70%)	HPLC	[32]
	μg/mL	1.96 (l); ND (st)	freeze-drying	UAE: EtOH (50%)	HPLC	[41]
**Caffeoylquinic acid**	X	X (l)	air-dried	EUA: MeOH (60%), 35 °C, 1 h; alkaline hydrolisis	HPLC-DAD-ESI-MS	[40]
	g/kg	8.47 (l + st)	fresh	Water (1:3 *w*/*w*); 100 °C; 60 min	HPLC-DAD	[36]
		9.07 (l + st)		EE: water (1:3 *w*/*w*), cellulose (0.01%)		
		1.64 (l + st)		UAE: Water (1:3 *w*/*w*); 75 °C; 60 min		
**5-caffeoylquinic acid**	mg/kg DW	4.30 (st) (P)	freeze-drying	EtOH (50%); 20 min; 75 °C	HPLC-PAD-ESI-MSn	[43]
		1.89 (st) (P)	oven-drying (40 °C, 72 h)			
		0.46 (st) (P)	temperature gradient (75–60 °C)			
	mg/kg DW	18.3 (st); 24.4 (rf)	freeze-drying	UAE: MeOH (100% *v*/*v*); 75 °C, 15 min	UPLC-MS/MS	[10]
**Ferulic acid**	mg/g Extract	0.24 (l); 0.58 (s)	dried oven (40 °C)	EtOH (70%)	HPLC	[32]
	μg/mL	23.85 (l); 21.92 (st)	freeze-drying	UAE: EtOH (50%)	HPLC	[41]
**Chlorogenic acid**	μg/mL	2.15 (l); 0.87 (st)	freeze-drying	UAE: EtOH (50%)	HPLC	[41]
**O-diglucoside**	X	X (l)	air-dried	EUA: MeOH (60%), 35 °C, 1 h; alkaline hydrolisis	HPLC-DAD-ESI-MS	[40]
**trisinapoyl-diglucoside**	X	X (l)				
**Sinapic acid**	mg/kg DW	21.3 (st); 54.3 (rf)	freeze-dried	UAE: MeOH (100% *v*/*v*); 75 °C, 15 min	UPLC-MS/MS	[10]
**Di-sinapoyl-diglucose**	mg/kg DW	3.89 (st) (P)	freeze-drying	EtOH (50%); 20 min; 75 °C	HPLC-PAD-ESI-MSn	[43]
		9.07 (st) (P)	oven-drying (40 °C, 72 h)			
		14.93 (st) (P)	temperature gradient (75–60 °C)			
	X	X (l)	air-dried	EUA: MeOH (60%), 35 °C, 1 h; alkaline hydrolisis	UPLC-MS/MS	[40]
**p-coumaroylquinic acid**	mg/kg DW	0.55 (P) (st)	freeze-drying	EtOH (50%); 20 min; 75 °C	HPLC-PAD-ESI-MSn	[43]
		0.38 (P) (st)	oven-drying (40 °C, 72 h)			
		0.09 (P) (st)	temperature gradient (75–60 °C)			
**Organic acids**						
**Malic acid**	mg/g (DW)	69.28 (sp)	freeze-dried	Water (100 °C; 1 h)	HPLC-DAD	[42]
	X	X (l)	dried (50 °C, 24 h)	DM: EtOH (30%); 50 °C; 36, 92 min	GC-EI-MS	[15]
**Citric acid**	mg/g (DW)	831.06 (sp)	freeze-dried	Water (100 °C; 1 h)	HPLC-DAD	[42]
**Oxalic acid**	mg/g (DW)	139.62 (sp)	freeze-dried	Water (100 °C; 1 h)	HPLC-DAD	[42]
**Gluconic acid**	UAA	141.67 (l); 84.22 (st); 420.64 (if)	air oven (45 °C, 48 h)	EUA: EtOH (80%), 45 °C, 1 h	HPLC-ESI-MS	[11]
**Cinnamic acid**	Peak area	102,226 (l)	freeze-dried	-	UHPLC-QTOF-MS	[33]
**Succinic acid**	X	X (l)	dried (50 °C, 24 h)	DM: EtOH (30%); 50 °C; 36, 92 min	GC-EI-MS	[15]
**Fumaric acid**	X	X (l)	dried (50 °C, 24 h)	DM: EtOH (30%); 50 °C; 36, 92 min	GC-EI-MS	[15]

Ref: References. By-products: (l) leaves; (st) stems/stalcks; (if) inflorescences; (sp) sprouts; (s) seeds; (rf): remains florets. P: Parthenon; N: Naxos; B: Beneforte; DW: dry weight; FW: fresh weight; ND: not detected; GAE: gallic acid equivalent; CE: catechin equivalent; QE: quercetin equivalent; RE: rutin equivalent; UAE: ultrasonic-assisted extraction; SFE: supercritical fluid extraction; MW: microwave; EE: enzymatic extraction; DM: dynamic maceration; UAA: arbitrary units of area; K: kaempferol; Q: quercetin; X: not quantified.

**Table 2 foods-13-03918-t002:** Glucosinolates and their breakdown products present in broccoli by-products.

Compound	Units	By-Products Content	Vegetable Processing	Extraction Method	Analysis Method	Ref.
**Glucosinolates and Their Breakdown Products**						
* Glucosinolates *						
Total glucosinolates	mg/g DW	5.00 (l + st)	freeze-drying	MeOH (70%), 70 °C, 30 min	UHPLC-DAD-ESI/MS	[39]
		5.15 (l + st)	air-drying (40 °C)			
		4.93 (l + st)	blanching and freeze-driying			
		2.50 (l + st)	blanching and air-drying (40 °C)			
		7.85 (l + st + if)	freeze-drying			
		6.06 (l + st + if)	MW hydrodiffusion and gravity			
	μmol/g DW	10.08 (l) (Gm); 7.45 (st) (Gm)	freeze-drying	MeOH (70%); 10 min; 95 °C (i.s. glucosinalbin)	UHPLC-PAD	[8]
	mg/kg DW	1836.6 (l); 5775.6 (rf)	freeze-drying	UAE: MeOH (100%), 75 °C, 15 min	UPLC-MS/MS	[10]
*Aliphatic glucosinolates*						
Total aliphatic glucosinolates	μmol/g FW	54.5–218.7 (s)	fresh	H_2_O, 100 °C, 15 min	HPLC	[45]
	mg/30 g FW	80.50 (sp)	freeze-drying	EtOH; methyl jasmonate (250 μM)	LC-MS	[21]
	mg/100 g DW	48 (st) (P)	freeze-drying	MeOH–Water–Formic acid (79/19/1 *v*/*v*/*v*)	HPLC-DAD-ESI-MS	[34]
Glucoraphanin	UAA	307.29 (l); 139.27 (st); 3149.91 (if)	air oven (45 °C, 48 h)	UAE: EtOH (80%), 45 °C, 1 h	HPLC-ESI-MS	[11]
	μmol/g DW	2.77 (l) (Gm); 3.79 (st) (Gm)	freeze-drying	MeOH (70%); 10 min; 95 °C (i.s. glucosinalbin)	UHPLC-PAD	[8]
	mg/kg DW	2633.89 (st) (P)	freeze-drying	EtOH (50%), 75 °C, 20 min	HPLC-PAD-ESI-MS	[43]
		2212.52 (st) (P)	oven-drying (40 °C, 72 h)			
		2157.00 (st) (P)	temperature gradient (75–60 °C)			
	μmol/g FW	2.5–129.9 (s)	fresh	H_2_O, 100 °C, 15 min	HPLC	[45]
	mg/kg DW	1586.6 (st); 5076.3 (rf)	freeze-drying	UAE: MeOH (100%), 75 °C, 15 min	UPLC-MS/MS	[10]
	μg/g DW	995–1136 (l) (P)	freeze-drying	PLE: EtOH (70%), 60 °C, 5 min, 1500 psi	UHPLC-QTOF-MS	[46]
		885–1012 (l) (Nb)				
		980–1090 (l) (Nx)				
	mg/30 g FW	51.08 (sp)	freeze-drying	EtOH; methyl jasmonate (250 μM)	LC-MS	[21]
	mg/100 g DW	45.0 (st) (P)	freeze-drying	MeOH–Water–Formic acid (79/19/1 *v*/*v*/*v*)	HPLC-DAD-ESI-MS	[34]
	mg/g DW	3.21 (st)	freeze-drying	MeOH (70%); 70°C, 30 min	HPLC-DAD-ESI-MS	[38]
		2.66 (st)	air-dried (60 °C)			
		3.73 (st)	air-dried (80 °C)			
Glucoiberin	μmol/g DW	0.65 (l) (Gm); 0.97 (st) (Gm)	freeze-drying	MeOH (70%); 10 min; 95 °C (i.s. glucosinalbin)	UHPLC-PAD	[8]
	mg/kg DW	403.02 (st) (P)	freeze-drying	EtOH (50%), 75 °C, 20 min	HPLC-PAD-ESI-MS	[43]
		566.64 (st) (P)	oven-drying (40 °C, 72 h)			
		407.99 (st) (P)	temperature gradient (75–60 °C)			
	μmol/g FW	0–65.9 (s)	fresh	H_2_O, 100 °C, 15 min	HPLC	[45]
	mg/kg DW	86.1 (st); 242.9 (rf)	freeze-drying	UAE: MeOH (100%), 75 °C, 15 min	UPLC-MS/MS	[10]
	μg/g DW	130–173 (l) (P)	freeze-drying	PLE: EtOH (70%), 60 °C, 5 min, 1500 psi	UHPLC-QTOF-MS	[46]
		110–130 (l) (Nb)				
		ND-11 (l) (Nx)				
	mg/30 g FW	19.28 (sp)	freeze-drying	EtOH; methyl jasmonate (250 μM)	LC-MS	[21]
Glucoerucin	μmol/g FW	3.8–108.4 (s)	Fresh	H_2_O, 100 °C, 15 min	HPLC	[45]
	mg/kg DW	245.2 (st); 45.0 (rf)	freeze-drying	UAE: MeOH (100%), 75 °C, 15 min	UPLC-MS/MS	[10]
	μg/g DW	ND-15/l) (P)	freeze-drying	PLE: EtOH (70%), 60 °C, 5 min, 1500 psi	UHPLC-QTOF-MS	[46]
		ND-17 (l) (Nb)				
		ND-<LOQ (l) (Nx)				
	mg/30 g FW	10.14 (sp)	freeze-drying	EtOH; methyl jasmonate (250 μM)	LC-MS	[21]
Gluconapin	μmol/g FW	0–16.2 (s)	fresh	H_2_O, 100 °C, 15 min	HPLC	[45]
	mg/kg DW	34.3 (st); 119.3 (rf)	freeze-drying	UAE: MeOH (100%), 75 °C, 15 min	UPLC-MS/MS	[10]
	μg/g DW	ND (l) (P, Nb, Nx)	freeze-drying	PLE: EtOH (70%), 60 °C, 5 min, 1500 psi	UHPLC-QTOF-MS	[46]
6(methylsulfonyl)hexylglucosinolate	X	X (l)	oven-drying (65 °C, 72 h)	EtOH (70%), 70 °C, 30 min	UHPLC-QTOF-MS	[47]
*Aromatic glucosinolates*						
Gluconasturtiin	μg/g DW	88–119 (l) (P)	freeze-drying	PLE: EtOH (70%), 60 °C, 5 min, 1500 psi	UHPLC-QTOF-MS	[46]
		51–68 (l) (Nb)				
		ND-30 (l) (Nx)				
Sinalbin	peak area	15672 (st)	freeze-drying	-	UHPLC-QTOF-MS	[33]
*Indolic glucosinolates*						
Total Indolic	μmol/g FW	0–7.2 (s)	fresh	H_2_O, 100 °C, 15 min	HPLC	[45]
	mg/30 g FW	40.62 (sp)	freeze-drying	EtOH; methyl jasmonate (250 μM)	LC-MS	[21]
	mg/100 g DW	20.0 (st) (P)	freeze-drying	MeOH–Water–Formic acid (79/19/1 *v*/*v*/*v*)	HPLC-DAD-ESI-MS	[34]
Glucobrassicin	μmol/g DW	0.24 (l) (Gm); 0.10 (st) (Gm)	freeze-drying	MeOH (70%); 10 min; 95 °C (i.s. glucosinalbin)	UHPLC-PAD	[8]
	mg/kg DW	109.81 (l) (P)	freeze-drying	EtOH (50%), 75 °C, 20 min	HPLC-DAD-ESI-MS	[43]
		113.27 (l) (P)	oven-drying (40 °C, 72 h)			
		57.92 (l) (P)	temperature gradient (75–60 °C)			
	μmol/g FW	0–3.8 (s)	Fresh	H_2_O, 100 °C, 15 min	HPLC	[45]
	mg/kg DW	100.0 (st); 868.1 (rf)	freeze-drying	UAE: MeOH (100%), 75 °C, 15 min	UPLC-MS/MS	[10]
	μg/g DW	85–102 (l) (P)	freeze-drying	PLE: EtOH (70%), 60 °C, 5 min, 1500 psi	UHPLC-QTOF-MS	[46]
		77–106 (l) (Nb)				
		26–38 (l) (Nx)				
	mg/30 g FW	9.38 (sp)	freeze-drying	EtOH; methyl jasmonate (250 μM)	LC-MS	[38]
	mg/100 g DW	0.0 (st) (P)	freeze-drying	MeOH–Water–Formic acid (79/19/1 *v*/*v*/*v*)	HPLC-DAD-ESI-MS	[34]
	mg/g DW	0.16 (st)	freeze-drying	MeOH (70%); 70 °C, 30 min	HPLC-DAD-ESI-MS	[38]
		0.47 (st)	air-dried (60 °C)			
		ND (st)	air-dried (80 °C)			
Methoxyglucobrassicin	UAA	199.97 (l); 202.39 (st); 2612.6 (if)	air oven (45 °C, 48 h)	UAE: EtOH (80%), 45 °C, 1 h	HPLC-ESI-MS	[11]
	mg/kg DW	449.05 (st) (P)	freeze-drying	EtOH (50%); 20 min; 75 °C	HPLC-PAD-ESI-MS	[43]
		416.89 (st) (P)	oven-drying (40 °C, 72 h)			
		327.92 (st) (P)	temperature gradient (75–60 °C)			
	mg/g DW	0.24 (st)	freeze-drying	MeOH (70%); 70 °C, 30 min	HPLC-DAD-ESI-MS	[38]
		0.06 (st)	air-dried (60 °C)			
		ND (st)	air-dried (80 °C)			
4-MGBS	μmol/g DW	0.17 (l) (Gm); 0.16 (st) (Gm)	freeze-drying	MeOH (70%), 95 °C, 10 min (i.s. glucosinalbin)	UHPLC-PAD	[8]
	mg/kg DW	294.8 (st); 431.0 (rf)	freeze-drying	UAE: MeOH (100%), 75 °C, 15 min	UPLC-MS/MS	[10]
	μg/g DW	15–35 (l) (P)	freeze-drying	PLE: EtOH (70%); 5 min; 60 °C; 1500 psi	UHPLC-qTOF-MS	[46]
		14–33 (l) (Nb)				
		ND-18 (l) (Nx)				
	mg/30 g FW	7.14 (sp)	freeze-drying	EtOH; methyl jasmonate (250 μM)	LC-MS	[21]
	mg/100 g DW	10 (st)(P)	freeze-drying	MeOH–Water–Formic acid (79/19/1 *v*/*v*/*v*)	HPLC-DAD-ESI-MS	[34]
* Isothiocyanate *						
Sulforaphane	mg/kg DW	1.09 (st) (P)	freeze-drying	EtOH (50%); 20 min; 75 °C	UHPLC-ESI-3Q-MS/MS	[43]
		0.35 (st) (P)	oven-drying (40 °C, 72 h)			
		0.62 (st) (P)	temperature gradient (75–60 °C)			
	mg/g FW	2.5–12.1 (s)	fresh	Acetone–Water–MeOH	HPLC	[9]
	mg/g seed	0–11.70 (sp) (3–7 day)	blanched (55 °C, 15 min)	Acetone–Water–MeOH	HPLC	[9]
	μmol/g FW	2199 (s)	fresh	HPH (5000 psi and 5 passes)	HPLC	[49]
	mg/g	7.5 (s)	fresh	MW pretreatment (3 min)	HPLC	[48]
		17.0 (s)		UAE (25–40 min; 500 W; 25–35 °C) Etyl acetate/water		
Indole-3-carbinol	mg/kg DW	1.50 (st) (P)	freeze-drying	EtOH (50%); 20 min; 75 °C	UHPLC-ESI-3Q-MS/MS	[43]
		3.95 (st) (P)	oven-drying (40 °C, 72 h)			
		LOQ (st) (P)	temperature gradient (75–60 °C)			
BITC	%	16.44 (l)	fresh	Maceration in DCM, 7 days	GC-MS	[50]
GER-ITC	μmol/g FW	2.2–67.1 (s)	fresh	H_2_O; 2 h; 25 °C; CH_2_Cl_2_ (i.s. ST-ITC)	GC-FID	[45]
GRA-ITC		2.6–91.1 (s)				
* Nitriles *						
Total nitriles	μmol/g FW	1.4–85.9 (s)	fresh	H_2_O; 2 h; 25 °C; CH_2_Cl_2_ (i.s. ST-ITC)	GC-FID	[45]
GER-CN		0–68.0 (s)				
GRA-CN		0–35.4 (s)				

Ref: References. By-products: (l) leaves; (st) stems/stalcks; (if) inflorescences; (sp) sprouts; (s) seeds; (rf): remains florets. Gm: Green magic; P: Parthenon; N: Naxos; Nb: Nubia; DW: dry weight; FW: fresh weight; ND: not detected; i.s.: internal standard; UAE: ultrasonic-assisted extraction; MW: microwave; DCM: dichloromethane; UAA: arbitrary units of area; HPH: high-pressure homogenization; PLE: pressurized liquid extraction; LOQ: still above the limit of quantification; X: not quantified; 4-MGBS: 4-Methoxyglucobrassicin; BITC: benzyl isothiocyanate; GER-ITC: 4-(methylthio)butyl isothiocyanate; GRA-ITC: 4-(methylsulfinyl)butyl isothiocyanate; GER-CN: 5-(methylthio)-pentanenitrile; GRA-CN: 5-(methylsulfinyl)pentanenitrile.

**Table 3 foods-13-03918-t003:** Pigments, vitamins, and minerals in by-products of broccoli.

Compound	Units	By-Products Content	Vegetable Processing	Extraction Method	Analysis Method	Ref.
**Pigmentts**						
* Chlorophyll *						
Total Chlorophyll	mg/g DW	0.25 (st)	MW–hot air-drying (40 °C)	Acetone (80%)	UV-Vis	[37]
		0.05 (st)	MW–hot air-drying (60 °C)			
		≈0.45 (st)	osmotic dehydration			
		0.96 (st)	fresh			
	mg/g	32.64 (l + st) (P)	dried (55 °C, 24 h)	SFE: CO_2_ 40 °C, 443 bar, 7% EtOH, 31 g/min of flow rate (68 min)	UV-Vis	[31]
		40.89 (l + st) (N)	dried (55 °C, 24 h)			
	mg/100 g DW	76.42 (l); 13.88 (st); 35.50 (if)	air oven (45 °C, 48 h)	Acetone (4 °C, 15 min)	UV-Vis	[11]
	μg/g DW	4103.7 (l + st)	freeze-drying	0.1% butylated hydroxytoluene—EtOH 100% (24 h)	UHPLC-DAD-ESI/MS	[39]
		1906.1 (l + st)	air-drying at 40 °C			
		5569.4 (l + st)	blanching and freeze-driying			
		1628.2 (l + st)	blanching and air-drying (40 °C)			
		175.3 (l + st + if)	freeze-drying			
		138.9 (l + st + if)	MW hydrodiffusion and gravity			
Chlorophyll a	mg/100 g DW	56.38 (l); 10.95 (st); 27.93 (if)	air oven (45 °C, 48 h)	Acetone (4 °C, 15 min)	UV-Vis	[11]
	μg/g DW	4477.9 (l) (Gm); 143.8 (st) (Gm)	freeze-drying	Acetone/MeOH (2:1 *v*/*v*, 0.5% BHT)	UHPLC	[8]
Chlorophyll b	mg/100 g DW	2.93 (l); 20.04 (st); 7.57 (if)	air oven (45 °C, 48 h)	Acetone (4 °C, 15 min)	UV-Vis	[11]
	μg/g DW	780.9 (l) (Gm); 22.2 (st) (Gm)	freeze-drying	Acetone/MeOH (2:1 *v*/*v*, 0.5% BHT)	UHPLC	[8]
* Carotenoids *						
Total Carotenoids	μg/g DW	1609.2 (l + st)	freeze-drying	0.1% butylated hydroxytoluene—EtOH 100% (24 h)	UHPLC-DAD-ESI/MS	[39]
		922.7 (l + st)	air-drying at 40 °C			
		2688.7 (l + st)	blanching and freeze-driying			
		963.0 (l + st)	blanching and air-drying at 40 °C			
		30.6 (l + st + if)	freeze-drying			
		15.6 (l + st + if)	MW hydrodiffusion and gravity			
	μg/g DW	1095.0 (l) (Gm); 15.6 (st) (Gm)	freeze-drying	Acetone/MeOH (2:1 *v*/*v*, 0.5% BHT); hexane (i.s. -apo-8′-carotenal)	UHPLC	[8]
β-carotene	μg/g DW	248.4 (l) (Gm); ND (st) (Gm)	freeze-drying		UHPLC	[8]
	mg/g DW	84.11 (l + st)(P)	dried (55 °C, 24 h)	SFE: CO_2_ 40 °C, 443 bar, 7% EtOH, 31 g/min of flow rate (68 min)	UV-Vis	[31]
		148.93 (l + st)(N)				
Lutein	μg/g DW	484.1 (l) (Gm); 10.8 (st) (Gm)	freeze-drying	Acetone/MeOH (2:1 *v*/*v*, 0.5% BHT); hexane (i.s. -apo-8′-carotenal)	UHPLC	[8]
Violaxanthin	μg/g DW	206.3 (l) (Gm); ND (st) (Gm)				
**Vitamins**						
* Vitamin C *	mg AA/g DW	2.02 (l); 2.69 (s)	dried oven (40 °C)	MeOH (70%)	UV-Vis	[32]
		2.31 (l); 2.25 (s)		EtOH (70%)		
		2.74 (l); »1.25 (s)		Hot water		
	mg/g DW	1.99 (st)	osmotic dehydration	Distilled water and 10% trichloroacetic acid, ice bath (5 min)	UV-Vis	[37]
		1.0 (st)	MW–hot air-drying (40 °C)			
		0.5 (st)	MW–hot air-drying (60 °C)			
		4.0 (st)	Fresh			
	g/30 g FW	5.45 (sp)	freeze-drying	-	-	[21]
	mg/g	1.08 (l); 0.6 (st)	freeze-drying	Oxalic acid/EDTA; HPO_3_-acetic; sulforic acid	UV-Vis	[33]
* Vitamin E *						
Total Tocopherol	μg/g DW	155.0 (l) (Gm); 1.97 (st) (Gm)	freeze-drying	Isopropanol/hexane (3:2 *v*/*v*), extracted with N_2_	UNPLC	[8]
α-Tocopherol	mg/g	5.69 (l + st) (P)	dried (55 °C, 24 h)	SFE: CO_2_ 40 °C, 443 bar, 7% EtOH, 31 g/min of flow rate (68 min)	GC-MS	[31]
		5.67 (l + st) (N)				
* Vitamin K *						
Phylloquinone	μg/g DW	24.3 (l) (Gm); 2.21 (st) (Gm)	freeze-drying	Isopropanol/hexane (3:2 *v*/*v*), extracted with N_2_	UHPLC	[8]
	μg/100 g	240 (st)	frozen (N_2_)	lipase treatment	LC-ESI-MS/MS	[52]
**Minerals**						
Mg	mg/100 g DW	364.67 (st)	dried (60 °C, 24 h)	HNO_3_:HCIO_4_ (2:1), 3–4 h. Sand bath 100 °C, HCl	AAS	[35]
	mg/g DW	1.33 (l) (Gm); 1.67 (st) (Gm)	freeze-drying	HNO_3_ (1 N), ashed 550 °C	ICP-AES	[8]
	mg/kg DW	2555.2 (st) (P)	freeze-drying	MW-assisted digestion (H_2_O_2_/HNO_3_ (1/4, *v*/*v*))	ICP-AES	[34]
Ca	mg/100 g DW	1245.34 (st)	dried (60 °C, 24 h)	HNO_3_:HCIO_4_ (2:1), 3–4 h. Sand bath 100 °C, HCl	AAS	[35]
	mg/g DW	28.99 (l) (Gm); 7.10 (st) (Gm)	freeze-drying	HNO_3_ (1 N), ashed 550 °C	ICP-AES	[8]
	mg/kg DW	4887.8 (st) (P)	freeze-drying	MW-assisted digestion (H_2_O_2_/HNO_3_ (1/4, *v*/*v*))	ICP-AES	[34]
Na	mg/100 g DW	63.54 (st)	dried (60 °C, 24 h)	HNO_3_:HCIO_4_ (2:1), 3–4 h. Sand bath 100 °C, HCl	AAS	[35]
	mg/g DW	2.63 (l) (Gm); 6.43 (st) (Gm)	freeze-drying	HNO_3_ (1 N), ashed 550 °C	ICP-AES	[8]
	mg/kg DW	2920.10 (st) (P)	freeze-drying	MW-assisted digestion (H_2_O_2_/HNO_3_ (1/4, *v*/*v*))	ICP-AES	[34]
Fe	mg/100 g DW	20.72 (st)	dried (60 °C, 24 h)	HNO_3_: HCIO_4_ (2:1), 3–4 h. Sand bath 100 °C, HCl	AAS	[35]
	mg/g DW	40.50 (l) (Gm); 15.83 (st) (Gm)	freeze-drying	HNO_3_ (1 N), ashed 550 °C	ICP-AES	[8]
	mg/kg DW	15.70 (st) (P)	freeze-drying	MW-assisted digestion (H_2_O_2_/HNO_3_ (1/4, *v*/*v*))	ICP-AES	[34]
Mn	mg/100 g DW	20.09 (st)	dried (60 °C, 24 h)	HNO_3_: HCIO_4_ (2:1), 3–4 h. Sand bath 100 °C, HCl	AAS	[35]
	mg/g DW	26.17 (l) (Gm); 7.00 (st) (Gm)	freeze-drying	HNO_3_ (1 N), ashed 550 °C	ICP-AES	[8]
	mg/kg DW	39.60 (st) (P)	freeze-drying	MW-assisted digestion (H_2_O_2_/HNO_3_ (1/4, *v*/*v*))	ICP-AES	[34]

Ref: References. By-products: (l) leaves; (st) stems/stalcks; (if) inflorescences; (sp) sprouts; (s) seeds. Gm: Green magic; P: Parthenon; N: Naxos; DW: dry weight; FW: fresh weight; i.s.: internal standard; SFE: supercritical fluid extraction; MW: microwave; BHT: butylated hydroxytoluene; EDTA: ethylenediaminetetraacetic acid; AA: ascorbic acid; AAS: atomic absorption spectroscopy; ICP-AES: inductively coupled plasma-atomic emissions spectroscopy.

**Table 4 foods-13-03918-t004:** Chemical compositions of diverse broccoli by-products (proteins, lipids, carbohydrates, and other).

Compound	Units	By-Products Content	Vegetable Processing	Extraction Method	Analysis Method	Ref.
**Protein**						
Total protein	% (*w*/*w*) DW	23.2 (pureed l + st)	freeze-drying	-	Leco TruMac Nitrogen analyser	[54]
		19.7 (pomace l + st)		-		
		25.4 (juice l + st)		-		
	g/30 g FW	0.86 (sp)	fresh	-	-	[21]
	%	6.13 (l)	fresh	Alkaline extraction + isoelectric precipitation (HCl)	Kjeldahl method	[53]
	g/100 g DW	22.75 (l); 14.1 (st)	freeze-drying	-	Kjeldahl method	[33]
	g/100 g DW	14.37 (st)	freeze-drying	-	Kjeldahl method	[21]
		13.36 (st)	air-dried (60 °C)	-		
		12.76 (st)	air-dried (80 °C)	-		
Crude protein	g/100 g DW	21.26 (l) (P); 19.44 (l) (P); 28.67 (if) (P)	air oven (45 °C, 48 h)	-	Kjeldahl method	[12]
Albumin	%	31.66 (l)	fresh	Water–NaCl (5%)–NaOH–EtOH (70%)	Enzymatic–Gravimetric method	[53]
Globulin	%	15.89 (l)				
* Amino acid *						
Lysine	X	X (s)	oven-dried	EtOH (70%, 1:20 *w*/*v*) orbital shaker, 50 °C (1 h)	UHPLC-QTOF-MS/MS	[26]
L-histidine	X	X (s)				
D-serine	X	X (sp + s)				
L-asparagine	X	X (sp + s)				
DL-o-tyrosine	X	X (s)				
L-Leucine	X	X (s)	dred (50 °C, 24 h)	DM: EtOH (30%); 50 °C, 36.92 min	GC-EIMS	[15]
L-phenylalanine	X	X (sp + s)	oven-dried	EtOH (70%, 1:20 *w*/*v*) orbital shaker, 50 °C (1 h)	UHPLC-QTOF-MS/MS	[26]
L-tryptopha	X	X (s)				
Glycitin	X	X (s)				
**Lipids**						
Total fatty acid	g/100 g DW	3.34 (l); 3.17 (st); 2.48 (if)	air oven (45 °C, 48 h)	-	Kjeldahl method	[12]
	% (*w*/*w*) DW	8.1 (pureed l + st)	freeze-drying	-	Australian Standard 2300 Method	[54]
		7.3 (pomace l + st)				
		7.8 (juice l + st)				
	g/30 g FW	0.13 (sp)	Fresh	-	AOAC method	[53]
	g/100 g DW	0.89 (st)	freeze-drying	Soxhlet	-	[38]
		1.34 (st)	air-dried (60 °C)			
		0.93 (st)	air-dried (80 °C)			
Palmitic acid	UAA	68.38 (l); 159.23 (st); 633.04 (if)	air oven (45 °C, 48 h)	UAE: EtOH (80%); 45 °C, 1 h	HPLC-ESI-MS	[11]
Linolenic acid	UAA	40.39 (l); 19.49 (st); 334.43 (if)				
8,15-DiHETE	peak area	82,203 (st)	freeze-drying	-	UHPLC-QTOF-MS	[33]
**Carbohydrates**						
Total carbohydrates	% (*w*/*w*) DW	55.7 (pureed l + st)	freeze-drying	-	Weight difference	[54]
		65.1 (pomace l + st)				
		45.2 (juice l + st)				
	g/30 g FW	0.09 (sp)	Fresh	-	Weight difference	[21]
	%	43.9 (l) (P)	freeze-drying	-	Weight difference	[34]
	g/100 g DW	82.27 (l)	freeze-drying	-	Weight difference	[38]
		76.43 (l)	air-dried (60 °C)	-		
		76.85 (l)	air-dried (80 °C)	-		
* Fiber *						
Total dietary fiber (TDF)	g/100 g DW	62.22 (l); 77.28 (st); 64.42 (if)	air oven (45 °C, 48 h)	AIR method (EtOH)	Enzymatic–Gravimetric method	[12]
	% (*w*/*w*) DW	36.5 (pureed l + st)	freeze-drying	-	Megazume fiber kit	[54]
		49.6 (pomace l + st)				
		2.0 (juice l + st)				
	g/30 g FW	0.74 (sp)	fresh	-	-	[21]
	% DW	26.0–32.6 (l) (B)	air-dried	Uppsala method	-	[40]
	g/100 g DW	38.0 (st)(P)	fresh	EtOH (80%); 70 °C, 30 min	Enzymatic–Gravimetric method	[34]
	g/100 g DW	17.71 (st)	freeze–drying	–	Enzymatic–Gravimetric method	[21]
		22.87 (st)	air–dried (60 °C)			
		22.34 (st)	air–dried (80 °C)			
Insoluble dietary fiber (IDF)	g/100 g DW	56.27 (l) (P); 66.18 (st) (P); 58.36 (if) (P)	air oven (45 °C, 48 h)	–	Enzymatic–Gravimetric method	[12]
	g/100 g DW	32.62 (st)	freeze–drying	α–amylase and amyloglucosidase (EtOH–acetone)	Enzymatic–Gravimetric method	[55]
	% DW	23.3–30.6 (l) (B)	air–dried	Uppsala method	-	[40]
	g/100 g DW	34.9 (st) (P)	fresh	EtOH (80%); 70 °C, 30 min	Enzymatic–Gravimetric method	[34]
Soluble dietary fiber (SDF)	g/100 g DW	5.94 (l) (P); 11.10 (st) (P); 6.06 (if) (P)	air oven (45 °C, 48 h)	–	Enzymatic–Gravimetric method	[12]
	g/100 g DW	3.19 (st)	freeze-drying	α-amylase and amyloglucosidase (EtOH-acetone)	Enzymatic–Gravimetric method	[55]
	% DW	1.8–2.3 (l) (B)	air-dried	Uppsala method	-	[40]
	g/100 g DW	3.2 (st) (P)	fresh	EtOH (80%); 70 °C, 30 min	Enzymatic–Gravimetric method	[34]
* Free Sugar *						
Total sugar	mg/g	»18 (l); »27 (st)	freeze-drying	EtOH-HCl	UV–Vis	[33]
Total soluble sugars (TSSs)	g/100 g DW	25.39 (l) (P); 18.58 (st) (P); 24.61 (if) (P)	air oven (45 °C, 48 h)	Distilled water	HPLC-RID	[12]
	% (proteins fraction)	40.31 (l)	fresh	-	Percentage difference	[53]
Uronic acid	%	30.1 (st) (P)	freeze-drying	-	Colorimetrit method	[34]
Uronic acid (TDF)	mg/g fiber	657.60 (l) (P); 634.11 (st) (P); 636.18 (if) (P)	air oven (45 °C, 48 h)	Hydrolysis (H_2_SO_4_)	HPLC-RID	[12]
	%	49.3 (st) (P)		-	Colorimetrit method	[34]
Uronic acid (SDF)	% DW	1.0–1.4 (l) (B)	air-dried	-	AOAC method	[40]
Uronic acid (IDF)	% DW	7.3–8.1 (l) (B)	air-dried	-	AOAC method	[40]
	%	40.6 (st) (P)	freeze-drying	-	Colorimetrit method	[34]
Glucose	%	20.6 (st) (P)	freeze-drying	-	GLC-FID	[34]
Glucose (TDF)	mg/g fiber	7.45 (l) (P); 9.24 (st) (P); 5.90 (if) (P)	air oven (45 °C, 48 h)	Hydrolysis (H_2_SO_4_)	HPLC-RID	[12]
	%	3.6 (st) (P)		-	Colorimetrit method	[34]
Glucose (SDF)	% DW	0.7–1.1 (l) (B)	air-dried	-	AOAC method	[40]
Glucose (IDF)	% DW	1.0–12.2 (l) (B)	air-dried	-	AOAC method	[40]
	%	4.0 (st) (P)	freeze-drying	-	GLC-FID	[34]
Galacturonic acid	% (in pectin fraction)	74.7 (st)	boiled under reflux (EtOH)	HNO_3_ (25 v/p); 30 min	HPLC-PAD	[1]
Rhamnose		5.4 (st)			GC-MS	
	%	1.8 (st) (P)	freeze-drying	-	GLC-FID	[34]
Rhamnose (TDF)	%	1.7 (st) (P)	freeze-drying	-	GLC-FID	[34]
Rhamnose (SDF)	% DW	4.1–5.9 (l) (B)	air-dried	-	AOAC method	[40]
Rhamnose (IDF)	% DW	0.7 (l) (B)	air-dried	-	AOAC method	[40]
	%	2.3 (st) (P)	freeze-drying	-		[34]
Galactose	%	10.0 (st) (P)	freeze-drying	-	GLC-FID	[34]
	% (in pectin fraction)	13.6 (st)	boiled under reflux (EtOH)	HNO_3_ (25 v/p); 30 min	GC-MS	[1]
Galactose (TDF)	%	11.4 (st) (P)	freeze-drying	-	GLC-FID	[34]
Galactose (SDF)	% DW	2.5–3.1 (l) (B)	air-dried	-	AOAC method	[40]
Galactose (IDF)	% DW	1.2–1.5 (l) (B)		-		
	%	10.7 (st) (P)	freeze-drying	-	GLC-FID	[34]
**Hydrocarbons**						
3-methyloctacosane	%	58.76 (l)	fresh	Hexane	GC-MS	[50]
15-methyltriacontane	%	15.09 (l)				
3,3,17,17-tetraethylnonadecane	%	33.71 (l)	fresh	Maceration in DCM 7 days	GC-MS	[50]
13-methylnonacosane	%	13.38 (l)				

Ref: References. By-products: (l) leaves; (st) stems/stalcks; (if) inflorescences; (sp) sprouts; (s) seeds. P: Parthenon; B: Beneforte; DW: dry weight; FW: fresh weight; UAE: ultrasonic-assisted extraction; UAA: arbitrary units of area; X: not quantified; DCM: dichloromethane.

**Table 5 foods-13-03918-t005:** Beneficial effects of different broccoli by-products.

Plant Material (Information)	Extraction Method	In Vivo/In Vitro	Analysis Method	Major Results	Ref.
*Antioxidant activity*					
Dried seeds	EtOH (70%, 1:20 *w*/*v*) orbital shaker, 50 °C (1 h)	In vitro	DPPH, ABTS	Broccoli seed extract had the lowest DPPH radical scavenging activity, with an inhibitory value of 31.34%, which was not significantly different from the values for broccoli sprout (33.32%).	[26]
Dried sprouts					
Dried stems	Acetone (50%)	In vitro	DPPH	Moderate activities were observed in stem samples (53.34%).	[35]
Dried leaves and stems (P and N)	SFE: CO_2_ 40 °C, 443 bar, 7% eToh, 31 g/min of flow rate (68 min)	In vitro	ABTS, TEAC	The highest value of antioxidant activity was obtained for the Naxos cultivar (338.69 ± 31.95 mg Trolox/g versus 262.57 ± 40.57 mg Trolox/g in the Parthenon cultivar).	[31]
Dried leaves, stems and inflorescences (TSX 007, M, BRO 2047, Pa, SP)	EtOH (80% *v*/*v*) ultrasound bath (1 h) 45 °C	In vitro	DPPH, ABTS	The leaves of varieties “Parthenon” and “Summer Purple” showed the greatest differences in DPPH values between the studied plant parts.	[11]
Dried leaves and seeds	EtOH (70%)	In vitro	DPPH, reducing power absorbance, ABTS	The reducing power and ABTS assays indicated that leaves achieved the strongest antioxidant capacity in comparison with that of seeds, while the highest radical scavenging activity was found in seeds via the DPPH assay.	[32]
Pureed stems and leaves	MeOH (80%); 50 min; 60 °C	In vitro	ORAC	The juice (441 μmol TE/g DW) fractions had much greater ORAC than the powdered broccoli stems and leaves (311 μmol TE/g DW) and powdered pomace (204 μmol TE/g DW).	[54]
Pomace (stems and leaves)					
Juice (stems and leaves)					
Seeds and sprouts (5 cultivars)	Gastrointestinal digestion	In vitro	DPPH, FRAP	After digestion, sprouts held 44.0% and 67.9% higher contents of TP and TF, as well as 54.9% and 31.7% higher values of DPPH and FRAP than seeds, respectively.	[9]
Stalks (Parthenon)	MeOH–Water–Formic acid (79/19/1 *v*/*v*/*v*)	In vitro	FRAP; ORAC	The values were 264.0 μmol TE/g DW and 2821.7 μmol TE/g DW, respectively.	[34]
Total fiber fraction (stalks)	MeOH–Water–Formic acid (79/19/1 *v*/*v*/*v*)	In vitro	FRAP; ORAC	The values were 102.7 μmol TE/g DW and 1666.9 μmol TE/g DW, respectively.	
Insoluble fiber fraction (stalks)	MeOH–Water–Formic acid (79/19/1 *v*/*v*/*v*)	In vitro	FRAP; ORAC	The values were 229.6 μmol TE/g DW and 1856.4 μmol TE/g DW, respectively.	
Leaves and stalks (fresh)	UAE: MeOH (0.125% formic acid), 20 °C; 15 min	In vitro	POD, APX, CAT, SOD, O^2−^, H_2_O_2_, DPPH, and ABST	POD activity (U/g) and APX (U/g) were significantly higher in stalks compared to leaves. The O^2−^ production rate (U/g) and H_2_O_2_ content (μmol/g) were both considerably lower in the stalks when compared to the leaves. The ABTS and DPPH radical scavenging rates (%) were observed to be higher in broccoli leaves.	[33]
Stalks (Spain) (freeze-dried)	EtOH (70%; 25 °C; 30 min)	In vitro	DPPH	Drying had a significant impact on this parameter, being higher in flours dried at 80 °C, followed by FD and finally air-drying at 60 °C.	[38]
Stalks (Spain) (air-dried 60 °C)					
Stalks (Spain) (air-dried 80 °C)					
Leaves and stems	Water (1:3 *w*/*w*); 100 °C; 60 min	In vivo	Electrical impedance (*Saccharomyces* and a *Saccharomyces* subjected to oxidative stress (H_2_O_2_ 1 mmol/L))	The Saccharomyces strain had a growth of 10.00 h. When broccoli extract EE is added to these conditions, the growth is 8.25 h. If broccoli extract is obtained by AE, growth is 8.30 h; compared to broccoli extract obtained by UAE, this reduces the time to 9.36 h.	[36]
	EE: water (1:3 *w*/*w*), cellulose enzyme (0.01%)				
	UAE: water (1:3 *w*/*w*); 75 °C; 60 min				
Broccoli residues	MeOH	In vitro	DPPH	The highest antioxidant scavenging activity appeared with methanolic extract (15.39 IC50 μg/mL).	[57]
	Water				
Leaves and stalks (fresh)	Maceration: hexane, chloroform, water	In vitro	DPPH, FRAP	DPPH (500 μg): leaf extract 52%; stalk extract 86%	[58]
				FRAP (500 g): leaf extract 52%; stalk extract 63.8%	
Stalks (fresh)	EtOH (70%) and water	In vitro	DPPH, ABTS, FRAP	The ethanolic extract of broccoli by-products had the highest antioxidant capacity as determined by DPPH radical scavenging activity (91.52 ± 1.59 mg TE/100 g) and FRAP (70.70 ± 2.30 mg TE/100 g) results.	[56]
*Antimicrobial activity*					
Stems (dried)	Acetone (50%)	In vitro (paper disk diffusion method)	Antibacterial assay (*Klebsiella* sp, *E. coli*, *S. aureus*, and *P. aeruginosa*)	The acetone extracts of broccoli stem and floret were effective against all the bacteria tested, especially against *Klebsiella* spp.	[35]
Leaves (fresh)	Hexane extracts	In vitro (agar well diffusion method)	Antimicrobial activity (2 g-positive bacteria (*B. subtilis* and *S. aureus*), 3 g-negative bacteria (*E. coli*, *Klebsiella pneumoniae*, and *Helicobacter pylori*), and 2 fungi (*Candida albicans* and *Aspergilus niger*))	The hexane extract showed strong activity on *E. coli* and *S. aureus* (with MIC = 62.5 and 31.125 μg/mL, respectively), and moderate activity on *B. subtilis*, *K. pneumoniae*, and *C. albicans* (with a MIC = 15.62 μg/mL) for the three strains.	[50]
	Maceration in DCM 7 days			DCM extract showed strong activity against *S. aureus* and *E. coli* (MIC =1.38 and 1.36 μg/mL, respectively), and moderate activity on *B. subtilis*, *K. pneumoniae*, and *C. albicans* (MIC = 3.9, 62.5 and 7.8 μg/mL, respectively). Both extracts showed low activity against *A. niger* with an activity index of 0.57 and 0.66, respectively.	
Leaves, stems, and inflorescences (dried) (TSX 007, M, BRO 2047, P, SP)	EtOH (80% *v*/*v*) ultrasound bath (1 h) 45 °C	In vitro	Antimicrobial activity (*B. cereus*, *S. aureus* and *L. innocua*)	The stem and inflorescence extracts, mainly from the “TSX 007” variety, showed a strong inhibitory effect on the three bacteria studied (81.3–94.2 %).	[11]
Leaves and seeds (dried)	EtOH (70%)	In vitro (well agar diffusion and MIC)	Antibacterial activity: Gram-negative bacteria (*S. aureus* and *B. subtilis*) and Gram-positive bacteria (*E. coli* and *S. typhimurium*)	Considerable antibacterial efficacy was observed in either leaf and seeds against *B. subtilis* and *Salmonella typhimurium* (0.39–0.78 mg/mL), along with weak activity against *S. aureus* and *E. coli* (1.56–3.13 mg/mL).	[32]
Sprouts	UAE: water (1 h + 20 min)	In vitro	Antimicrobial activity (*E. coli*, *Salmonella typhimurium*, *Listeria monocytogenes*, *B. cereus*, and *S. aureus*)	*E. coli* and *L. monocytogenes* were sensitive to broccoli extracts (5 and 7 mg/mL MICINT, respectively)	[42]
				Broccoli extracts were also the most effective against *S. aureus* (6 mg/mL MICINT), being the best antimicrobial activity against *B. cereus* (8 mg/mL MICINT).	
Leaves and stems	Water (1:3 *w*/*w*); 100 °C; 60 min	In vivo (electrical impedance)	Antimicobial activity: *Salmonella enterica*, *listeria*	When broccoli extract obtained by EE is added, the growth of the *Listeria* strain is completely inhibited, with this EE being the one with the highest antimicrobial capacity against the *Listeria* strain. This is because the freeze-dried broccoli extract obtained by EE has the highest concentration of chlorogenic acid.	[36]
	EE: water (1:3 *w*/*w*), cellulose enzyme (0.01%)				
	UAE: water (1:3 *w*/*w*); 75 °C; 60 min				
Leaves and stalks (fresh)	Maceration: water	In vitro (agar well diffusion and MIC)	*S. aureus*	MIC: 0.5 mg/mL leaf extract; 0.25 mg/mL stalks extract	[58]
				MBC: 0.5 mg/mL leaf extract; 0.2 mg/mL stalks extract	
				The stalk extract showed superior activity compared to the leaf extract.	
Stalks (Avenger)	Macerated in an electric juice extractor	In vitro (well diffusion assay)	Pathogenic bacteria (*B. cereus*, *S. xylosus*, *S. aureus*, *Shigella flexneri*, *Shigella sonnei*, *Proteus vulgaris*), phytopathogenic fungi (*C. gloeosporioides*, *A. niger*), and yeasts (*C. albicans* and *Rhodotorula* sp.)	Proteolytic enzymes had a reduction of approximately 60% in antibacterial activity against *Staph. Xylosus*. This effect is partially due to AMPs.	[60]
Peptides seeds (fresh) (Avenger)	Molecular exclusion techniques + HPLC-MS	In vitro (well diffusion assay)	Antifungal and antibacterial activities: *C. gloeosporioides*, *Alternaria alternata*, *B. cereus*, *Listeria monocytogenes*, *Salmonella typhimurium*, *P. aeruginosa*, *Vibrio parahaemolyticus*, *Bifidobacterium animalis*, *Lactobacillus acidophilus*, and *L. casei.*	Two peptides showed properties of defensins from Brassica napus (98–71%) and Arabidopsis thaliana (6–85%). The combination of both altered the membranes of *C. gloeosporioides* and *Alternaria alternata* and also reduced the hyphal growth of *C. gloeosporioides* (~56%, 120 h). Pathogenic bacteria were susceptible, but probiotic bacteria were not inhibited.	[59]
Leaves (dried)	Four extractions: EtOH; UAE + EtOH; ChCl-PG; UAE + ChCl-PG	In vitro (well agar diffusion and MIC)	*S. aureus*, *E. coli*, and *Salmonella* spp.	The extracts obtained using UAE + ChCl-PG treatment showed greater antimicrobial potential against *S. aureus*, *E. coli*, and *Salmonella* than those obtained using the other three treatments.	[61]
*Anticancer activity*					
Seed flours	Acetone (50%)	In vitro	LNCaP prostate cancer	After 48 h of treatment, the extract inhibited LNCaP prostate cancer cell proliferations.	[62]
				The antiproliferative capacity was 20.0% after 96 h.	
Sprouts (fresh)	EtOH (70%)	In vitro	MTT assay: A549 (lung carcinoma cells), HepG2 (hepatocellular carcinoma cells), and Caco-2 (colorectal adenocarcinoma cells)	Significant antiproliferative activities against lung (A549), liver (HepG2), and colon (Caco-2) cancer cells were observed, with IC50 values ranging from 0.117 to 0.189 mg/mL over 48 h. Additionally, Caco-2 cells underwent apoptosis and showed a loss of mitochondrial membrane potential.	[65]
Leaves and Seeds (Dried)	EtOH (70%)	In vitro	Cytotoxic activity: A549 (lung carcinoma cells), Caco-2 (colorectal adenocarcinoma cells), and HepG2 (hepatocellular carcinoma cells)	Seeds extracts exerted significant cytotoxicity against A549, Caco-2, and HepG2 cancer cell lines at low inhibitory concentration (IC) 50 values (0.134, 0.209, and 0.238 mg/mL, respectively).	[32]
Broccoli residues	MeOH	In vitro	Cytotoxic activity: cancer cell lines liver HepG2, breast MCF-7, and colon HCT116	Broccoli methanol extract displayed potent growth-inhibitory activity against cancer cell lines of liver HepG2, breast MCF-7, and colon HCT 116. The remarkable cytotoxic activity of methanolic extract could be a result of its high contents of phenolics, flavonoids, and tannins.	[57]
	Water				
eCO2-treated sprouts (Southern star, Prominence, Monotop)	MeOH (70%); 70 °C; 30 min	In vitro	Induction of QR in Hepa lclc7 cells; measuring of glutathione-S-transferase	Broccoli sprout cultivars at high levels of CO_2_ significantly increased (x3) (*p* < 0.05) their levels of QR and GST anticancer indicators.	[68]
Sprouts (fresh)	MeOH (80%) extract fractioned (hexane, chloroform, ethyl acetate, butanol, water)	In vitro	MTT assay: breast cancer stem cells (BCSCs)	The chloroform fraction and hexane fraction reduced the viability of BCSCs, with chloroform being the most effective (IC50 = 69.47 mg/mL).	[63]
Peptides hydrolysates from broccoli stems	Water (1:2 p/w) hydrolysis (tripsin 50 °C, 4 h)	In vitro	MTT assay: HaCaT keratinocyte cell line	Concentrations of E and MF (20–10 lg/mL) had a significant negative effect on cell proliferation, decreasing cell viability down to 60–80% in the first 24 h when compared with control, untreated cell culture conditions.	[64]
*Anti-inflammatory activity*					
Seed flours	Acetone (50%)	In vitro	LPS-stimulated J774A.1 cells	Inhibition of 38.9% on IL-1β mRNA expression was observed when compared to the LPS-induced control.	[62]
				Broccoli seed flour extract showed a significant inhibition on COX-2 mRNA expression, with an inhibition rate of 33.8%.	
Sprouts (fresh)	-	In vivo (40 healthy overweight subjects, 30 g/day; 10 weeks)	Levels of TNF-a, IL-6, IL-1b, and C-reactive protein	IL-6 levels significantly decreased (mean values from 4.76 pg/mL to 2.11 pg/mL with 70 days) and during the control phase, the inflammatory levels were maintained at a low grade (mean values from 1.20 pg/mL to 2.66 pg/mL).	[71]
eCO_2_-treated sprouts (Southern star, Prominence, Monotop)	MeOH (70%); 70 °C; 30 min	In vitro	Lipoxygenase (LOX) assay; evaluation of cyclooxygenase-1 and cyclooxygenase-2	The eCO_2_ treatment inhibited COX-2 and LOX activities in broccoli sprouts extracts by increasing sulforaphane, indicating high anti-inflammatory potentiality.	[68]
Leaves and stalks (fresh)	Maceration: hexane, chloroform, water	In vitro	Inhibition of albumin denaturation	The stalk and leaf extracts of a plant showed superior inhibition of albumin denaturation, with 46% and 42%, respectively, outperforming the aspirin standard (37%).	[58]
			Antiproteinase action	The maximum inhibition was recorded at 500 μg/mL (71% for leaf extract and 78% for stalk extract), surpassing the aspirin standard’s 62% inhibition.	
			Membrane stabilization	The maximum inhibition was recorded at a concentration of 500 μg/mL with leaf sample inhibiting around 58% and the stalk extract inhibiting around 95%.	
Sprouts (fresh)	phosphate-buffered saline (4 °C, 30 min)	In vitro	ELISA	The extract significantly decreased the formation of AGEs in in vitro tests.	[66]
			RT-PCR: MCP-1, RAGE, ICAM-1, eNOS	In endothelial cells exposed to TNF-α and AGEs, the extract reduced the expression of proinflammatory genes (MCP-1, ICAM-1, RAGE). An increase in eNOS expression was observed.	
			ROS generation	The extract decreased the production of ROS in the cells, indicating protection against oxidative damage.	
Leaves	MeOH (70%)	In vivo (30 male rats)	RT-PCR: TNF-α, IL-1β, NF-kB	Treatment with broccoli leaf extract resulted in a significant reduction in TNF-α, IL-1β, and NF-kB expression in renal and hepatic tissues.	[69]
Leaves (dried)	EtOH (72 h)	In vitro	ELISA: PGE_2_ and cytokines (TNF-α, IL-6)	Treatment with the extract at 50 and 100 μg/mL decreased TNF-α secretion by 12% and 22%, as well as IL-6 secretion by 28% and 54%, respectively.	[67]
Sprouts (fresh)	EtOH (70%)	In vitro	MTT; NO assay; ELISA (IL-1β y TNF-α)	The extract significantly reduced NO production in a dose-dependent manner.	[70]
		In vivo (rats)	ALT and AST assay; survival analysis	Pretreatment with the extract resulted in lower ALT and AST levels, and a 50% mortality rate.	
			qPCR: IL-6, IL-1β, TNF-α, iNOS, COX-2	Treatment with BSE significantly reduced the expression of these inflammatory markers in the liver.	
			Western blot: iNOS, COX-2, y NF-κB		
*Anti-hipertensive activity*					
Peptides of stems and leaves proteins	Enzymatic hydrolysis (trypsin)	In vitro	ACE and DPP-IV inhibition assay	Two peptides were identified. LPGVLPVA exhibited an ACE IC 50 value of 0.776 μM and a DPP-IV IC 50 value of 392 μM. YLYSPAYshowed an ACE IC 50 value of 8.52 μM and a DPP-IV IC 50 value of 181 μM.	[72]
Peptides of stems and leaves proteins	Enzymatic hydrolysis (papain)	In vitro	ACEI activity assay	Two novel peptides, FVLPLR and LPWYR, were obtained. Their IC50 values for inhibiting ACE activity in vitro were 3.06 μM and 19.05 μM, respectively.	[73]
Activated and non-activated broccoli seeds extract (sulforaphane)	Commercial	In vivo (12 women with pregnancy hypertension and 6 non-pregnant women)	Plasma analytes (LC-MS)	The myrisinase-activated extract produced higher levels of sulforaphane	[74]
			Blood pressure	Sulforaphane showed a slight reduction in diastolic blood pressure and in circulating levels of the marker sFlt-1, linked to oxidative stress in preeclampsia.	
			ELISA (sFlt-1, PIGF, Activin A)		
Dried leaves, stems, and inflorescences (TSX 007, M, BRO 2047, P, SP)	EtOH (80% *v*/*v*) ultrasound bath (1 h) 45 °C	In vitro	ACE-inhibitory activity assay	Antihypertensive activity was only found in the extracts obtained from broccoli leaves, with the Parthenon variety showing the highest antihypertensive capacity.	[11]
*Effects on gut health*					
Seeds extract (SFN)	Soxhlet (MeoH 70%)	Two	16S rDNA sequencing analysis ITC, GSL, and polyphenols quantification	A significant increase was observed in beneficial bacteria such as *Bifidobacterium*, *Bilophila*, and *Coprococcus*. Significant inhibition of harmful bacteria such as Lachnoclostridium and Escherichia was observed.	[44]
Stalks (Parthenon)	EtOH (50%) (70 °C for 20 min)	In vitro	Trypan blue-based viability assay, citotoxicity (Caco-2)	The highest concentration of bioaccessible SFN was provided by the high-ITC materials (~4.00 mg/kg dw). The extracts tested showed no cytotoxic effects on Caco-2 cells and reduced the production of proinflammatory cytokines (IL-6, IL-8, and TNF-α) in the intestinal epithelium.	[76]
Sprouts	Fresh	In vivo (32 male rats, 7 weeks)	Gut microbiota analisys (16Sr RNA) ELISA, Western blot, RT-qPCR, gut permeability	Broccoli consumption reduced inflammation, as indicated by lower expression levels of iNOS and SAA1 in the broccoli-supplemented groups. Gut microbiota analysis revealed an increased abundance of Acidifaciens and a decreased abundance of Mucispirillum schaedleri in the stem-fed group.	[20]
Dietary fiber leaves	AIR method	In vitro	Microbial population; antiproliferative activity (HT-29)	Broccoli fiber promoted the growth of bifidobacteria in the colon, outcompeting lactic acid bacteria. It showed moderate prebiotic effects by stimulating SCFA production, with propionic acid being prominent. It exhibited significant antiproliferative activity in HT-29 cells.	[75]

Ref: References. N: Naxos; P: Parthenon; M: Monaco; SP: Summer Purple; TEAC: ascorbic acid equivale; ORAC: oxygen radical absorbance capacity; FRAP: ferric reducing antioxidant potential; QR: quinone reductase; DW: dry weight; TE: trolox equivalents; POD: peroxidase; APX: ascorbate peroxidase; CAT: catalase; SOD: superoxide dismutase; EE: enzymatic extraction; AE: aqueous extraction; UAE: ultrasonic-assisted extraction; DCM: dichloromethane; MIC: minimum inhibitory concentration; AMPs: antimicrobial peptides; E: peptides extracted from total protein; MF: peptides extracted from membrane protein; ROS: Reactive Oxygen Species; MTT: 3-(4,5-dimethylthiazol-2-yl)-2,5-diphenyltetrazolium bromide; RT-PCR: Real-Time Reverse Transcription–Polymerase Chain Reactions; GST: glutathion S-transferase; ACE: angiotensin I-converting enzyme; MCP-1: monocyte chemoattractant protein-1; ICAM-1: intercellular adhesion molecule-1; RAGE: receptor for advanced glycation end products; eNOS: upregulated endothelial nitric oxide synthase; TNF-α: tumor necrosis factor alpha: IL-1β: interleukin-1β; NFkB: nuclear factor kappa B: DPP-VI: dipeptidyl peptidase-IV; sFlt-1: soluble fms-like tyrsoine kinase-1; PIGF: placental growth factor; AIRs: alcohol-insoluble residues; iNOS: inducible nitric oxide synthase; SAA1: serum amyloid A1.

## Data Availability

Data are contained within the article.

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
