# Peer review of "Comprehensive Analysis of Bioactive Compounds, Functional Properties, and Applications of Broccoli By-Products"

_foods, 2024, doi:10.3390/foods13233918_

Round 1
Reviewer 1 Report
Comments and Suggestions for Authors
Comments on the manuscript entitled "Comprehensive Analysis of Bioactive Compounds, Functional Properties, and Applications of Broccoli By-Products" by Gudiño et al.
The authors provided a comprehensive review of the latest trends in extraction methods for nutrients and phytochemicals from broccoli by-products, along with a detailed analysis of the beneficial properties of these by-products, including leaves, stems, sprouts, and inflorescences. They described both positive edges and different challenges of incorporating these by-products into food products, with emphasis on enhancing total phenolic content and antioxidant capacity. Overall, the manuscript is well-written and analysed. However, minor corrections should be made before its acceptance:
1. Please change the term extract method to extraction conditions, extraction method, or method of extraction in Tables 1 to 5. Also, change the term analys method to analytical method or analysis method and make it consistent throughout the tables.

Author Response
Revisions made to the manuscript are highlighted in red so that editors and reviewers can see the changes made.
Reviewer 1
Comments and Suggestions for Authors
Comments on the manuscript entitled "Comprehensive Analysis of Bioactive Compounds, Functional Properties, and Applications of Broccoli By-Products" by Gudiño et al.
The authors provided a comprehensive review of the latest trends in extraction methods for nutrients and phytochemicals from broccoli by-products, along with a detailed analysis of the beneficial properties of these by-products, including leaves, stems, sprouts, and inflorescences. They described both positive edges and different challenges of incorporating these by-products into food products, with emphasis on enhancing total phenolic content and antioxidant capacity. Overall, the manuscript is well-written and analysed. However, minor corrections should be made before its acceptance:
- Please change the term extract method to extraction conditions, extraction method, or method of extraction in Tables 1 to 5. Also, change the term analys method to analytical method or analysis method and make it consistent throughout the tables.
AU: We appreciate the reviewer’s comments. The term “extract method” has been revised to “extraction method” in Tables 1 to 5. Additionally, the term “analys method” has been corrected to “analysis method”.

Reviewer 2 Report
Comments and Suggestions for Authors
Comments and suggestions for authors are in attachment.

Author Response
Revisions made to the manuscript are highlighted in red so that editors and reviewers can see the changes made.
Reviewer 2
General concept comments: The manuscript is of high relevance to the Foods topic. It is
very well structured and organized, it maintains its focus and has up to date references.
The proposed objectives are well defined and have been achieved.
AU: We appreciate the reviewer’s comment..
Specific comments:
As different units are being mentioned throughout the manuscript, make sure everything
is consistent, for instance:
Lines 130-134: The authors don't mention what are GAE. Should be mentioned as
it first appears in the manuscript. In table 1 you have these well explained in the footnote,
but it appears here first.
Line 158: Same comment for CE.
Line 162: Same comment for QE.
Line 207: Same comment for UAA.
AU: The meaning of the terms mentioned (GAE, CE, QE, UAA) have been explained in the text.
Lines 184-186: What previous study?
AU: According to the reviewer's comment, a reference has been added to clarify which article is being commented on.
Lines 190-191: What enzyme?
AU: We appreciate the reviewer's comment. The enzyme used in the study has been indicated in the text.
Regarding the tables: I sometimes find it confusing to correlate the respective row content
to each column (visual layout). Especially when you show different by-product contents
or different extraction methods for the same reference. Maybe have a thinner line (or use
intercalating colours for the rows) separating each table row by extract/reference. This
comment applies to all tables.
AU: According to the reviewer's comment, the tables have been modified to clarify the correlation of the data contained.
Lines 269-270: No need to write again fully (UAE) as you have already done it in
line 62.
AU: The method has been replaced by its acronym in the text.
Lines 302-303: Same as the previous comment but for SFE.
AU: As with the previous comment, the redaction has been modified.
Line 314: 0.6 mg/g – Is this FW?
AU: The authors of the study from which the data were extracted did not specify whether the vitamin C units, expressed in mg/g, are based on fresh weight (FW) or dry weight (DW). Consequently, this review does not specify the unit type. However, the authors did subject the broccoli samples to freeze-drying during processing (DW).
Line 319: mg AA/s DW – same units as in line 314? It is a bit confusing.
AU: We appreciate the reviewer’s question in this issue .The units mgAA/g and mg/g represent the same thing in the analysis of vitamin C or ascorbic acid (AA). However, in this review, the data are presented according to the units used in the cited studies.
Line 322: In this section you present three different units - mg/g; mgAA/s DW;
mg/g DW. Isn't mgAAs the same as just mg?
AU: As with the previous comment, the units mentioned are equivalent when expressing vitamin C (ascorbic acid) content. Despite being the same, the original units used in each study cited have been maintained.
Line 480: Which by-products?
AU: The sentence has been modified to clarify byproducts analyzed in the cited study.
Line 487: Saccharomyces in Italic.
AU: We thank the reviewer´s comment. The word has been modified.
Line 496: In this section the authors don't mention any specific values in the text,
only this one. Why this one? Is it relevant? Otherwise, I suggest removing the value as
well. You have these in the table.
AU: The value has been removed from the text according to the reviewer's suggestion.
Line 604: You have mentioned this information before, no need to do so here. I
suggest just saying "Sulforaphane has also been...".
AU: We appreciate the reviewer’s comment. The sentence has been modified.
Table 5: Saccharomyces in Italic in major results.
AU: We thank the reviewer´s comment. The word has been modified.
Table 5: You already have the full species name written in section 4.2. The first
time you mention a species, the name should be fully written, afterwards use the
abbreviated form. For E. coli and S. aureus you write again fully, while for P. aeruginosa
you abbreviate. Be consistent in this table section.
AU: We appreciate the reviewer’s contribution. The name of species have been modified as suggested by the reviewer to correct the imprecisions previously written in the section and its table.
Line 660: I don't believe this is de adequate phrasing to describe what you want.
Maybe change "prevalence" to "diagnosis" or the phrase to "number of celiac disease
cases.".
AU: According to reviewer comment, the redaction has been modified to clarify the meaning of the sentence.
Line 706: Lantiplantibacillus plantarumin in Italic.
AU: According to reviewer comment, the word has been modified.
Conclusions: Addition of some future perspectives regarding this thematic should be
included.
AU: The Conclusions have been modified as suggested by the reviewer to addition of future research.

Reviewer 3 Report
Comments and Suggestions for Authors
This paper has reviewed analysis of nutritionally valuable compounds in various parts of broccoli based on processing methods and analytical techniques. Furthermore this paper reviewed beneficial effects of broccoli and also various applications. This paper seems to be informative and could be valuable for the readers. Especially some tables are well summarized and very informative.
However, several things should be corrected or be reconsidered before publication.
1. In the section of 3.8. lipids, authors seemed to be confused between lipids and fatty acids. In line 375, fatty acids should be changed to lipids.
2. Review paper should not just transfer data from articles. In the several cases, it is necessary to modify the data for comparison. For example, in case of total chlorophyll in table 3, mg/g DW, mg/100 g DW, μg/g DW were used. These data could be converted easily in the same unit, and then readers can compare the results easily. In case of Na, mg/100 g DW, mg/g DW, and mg/kg DW were used. These data should be converted into the same unit. In case of total protein in table 4, %(W/W) DW, g/100 g DW were used. In fact, they are the same unit. This should be corrected. I just showed some examples. The others like these examples should be corrected.
3. In the section 3.7.1 amino acids, amino acids should be written in free amino acids. However, the amino acid data in table 4 do not give any information. It may be better to remove this section.
4. In the section 3.9.2. sugars, contents are very confusing. Do sugars mean free sugars or sugars including polysaccharides? It is necessary to explain in detail including experimental methods if necessary.
5. In the section of beneficial effects, it will be better to include effects of broccoli on gut health and gut microbiome because gut health is quite important and many reports related to the effects of broccoli on gut health have been published. Furthermore, recently effects of phytochemicals on gut health become new area for investigation.
Author Response
Revisions made to the manuscript are highlighted in red so that editors and reviewers can see the changes made.
Reviewer 3
Comments and Suggestions for Authors
This paper has reviewed analysis of nutritionally valuable compounds in various parts of broccoli based on processing methods and analytical techniques. Furthermore, this paper reviewed beneficial effects of broccoli and also various applications. This paper seems to be informative and could be valuable for the readers. Especially some tables are well summarized and very informative.
However, several things should be corrected or be reconsidered.
- In the section of 3.8. lipids, authors seemed to be confused between lipids and fatty acids. In line 375, fatty acids should be changed to lipids.
AU: According to reviewer comment, the redaction has been modified to clarify the meaning of the sentence.
- Review paper should not just transfer data from articles. In the several cases, it is necessary to modify the data for comparison. For example, in case of total chlorophyll in table 3, mg/g DW, mg/100 g DW, μg/g DW were used. These data could be converted easily in the same unit, and then readers can compare the results easily. In case of Na, mg/100 g DW, mg/g DW, and mg/kg DW were used. These data should be converted into the same unit. In case of total protein in table 4, %(W/W) DW, g/100 g DW were used. In fact, they are the same unit. This should be corrected. I just showed some examples. The others like these examples should be corrected.
AU: We appreciate your observation regarding the unification of units to facilitate the comparison of data across the studies presented in the tables. We agree that this approach could make direct comparisons between articles easier. However, as this is a review paper, we believe it is more appropriate to keep the data in the original units reported by the authors in the cited studies. This ensures the integrity and fidelity of the information as it was presented in the primary sources.
- In the section 3.7.1 amino acids, amino acids should be written in free amino acids. However, the amino acid data in table 4 do not give any information. It may be better to remove this section.
AU: Thank you for your comment. We consider it necessary to include the information on free amino acids in section 3.7.1, since this is a comprehensive review of the studies on the compounds present in broccoli by-products. Table 4 provides a summary of the data available in the literature on amino acids, which, although does not provide new information, is essential to provide a complete overview of the compounds studied in this context. We believe that maintaining this section contributes to the review's objective of providing a comprehensive and detailed review.
- In the section 3.9.2. sugars, contents are very confusing. Do sugars mean free sugars or sugars including polysaccharides? It is necessary to explain in detail including experimental methods if necessary.
AU: We appreciate your comment and realize that section 3.9.2, on sugars, can be confusing. To clarify, in this study we refer to “sugars” as free sugars, which includes monosaccharides and disaccharides present in the samples. We have revised the wording to make this distinction clearer and included experimental methods, which focus on the quantification of these specific sugars.
- In the section of beneficial effects, it will be better to include effects of broccoli on gut health and gut microbiome because gut health is quite important and many reports related to the effects of broccoli on gut health have been published. Furthermore, recently effects of phytochemicals on gut health become new area for investigation.
AU: We appreciate the reviewer’s contribution. A new section has been added to section 4 on the beneficial effect on intestinal health.

Reviewer 4 Report
Comments and Suggestions for Authors
The paper investigatedthe application and value of Broccoli and its by-products in the food industry. The article highlights the potential of these by-products as natural additives, thickeners and preservatives, as well as their role in promoting healthier and sustainable food consumption. In addition, the article mentions that using these by-products can reduce agricultural waste and promote a circular economy. Overall, the article is well-organized and well-expressed. However, there are still some issues that need to be improved:
Q1:The abstract lacks a period at the end.
Q2:L130, there should be a space between the number and the unit, and the whole text should be modified in turn.
Q3:L186,the word “study” should be changed to “studies”.
Q4:L213, Table 1 headings should be centered, full text modification.
Q5:L213, the word “0,45” should be changed to “0.45”.
Q6:L213,the Phenolic acids in Table 1 should conform to the format of Coumarins above, and be modified in the following paragraphs.
Q7:L288, the format in Table 2 remains consistent,and the word “temperature gradient(75-60°)” should be changed to “temperature gradient(75-60℃)”.
Q8:L331-332, vitamin k related description is too little, please add.
Q9:L400, data sink error occurred in the Carbohydrates section in Table 2.
Q10:L488-490, this paragraph does not connect with the context.
Q11:L611, the format of the Major results section in Table 3 is consistent, all of which are in full-head or indented form.
Q12:L686-687, the location of reference 29 is incorrect, and “incorporated” is not capitalized at the beginning of a sentence
Q13:Information in references should be complete and formatted consistently.
Please revise them.
Comments on the Quality of English Language
The paper investigatedthe application and value of Broccoli and its by-products in the food industry. The article highlights the potential of these by-products as natural additives, thickeners and preservatives, as well as their role in promoting healthier and sustainable food consumption. In addition, the article mentions that using these by-products can reduce agricultural waste and promote a circular economy. Overall, the article is well-organized and well-expressed. However, there are still some issues that need to be improved:
Q1:The abstract lacks a period at the end.
Q2:L130, there should be a space between the number and the unit, and the whole text should be modified in turn.
Q3:L186,the word “study” should be changed to “studies”.
Q4:L213, Table 1 headings should be centered, full text modification.
Q5:L213, the word “0,45” should be changed to “0.45”.
Q6:L213,the Phenolic acids in Table 1 should conform to the format of Coumarins above, and be modified in the following paragraphs.
Q7:L288, the format in Table 2 remains consistent,and the word “temperature gradient(75-60°)” should be changed to “temperature gradient(75-60℃)”.
Q8:L331-332, vitamin k related description is too little, please add.
Q9:L400, data sink error occurred in the Carbohydrates section in Table 2.
Q10:L488-490, this paragraph does not connect with the context.
Q11:L611, the format of the Major results section in Table 3 is consistent, all of which are in full-head or indented form.
Q12:L686-687, the location of reference 29 is incorrect, and “incorporated” is not capitalized at the beginning of a sentence
Q13:Information in references should be complete and formatted consistently.
Please revise them.
Author Response
Revisions made to the manuscript are highlighted in red so that editors and reviewers can see the changes made. Three reviewers have already replied to the manuscript.
Revierwer 4
Comments and Suggestions for Authors
The paper investigatedthe application and value of Broccoli and its by-products in the food industry. The article highlights the potential of these by-products as natural additives, thickeners and preservatives, as well as their role in promoting healthier and sustainable food consumption. In addition, the article mentions that using these by-products can reduce agricultural waste and promote a circular economy. Overall, the article is well-organized and well-expressed. However, there are still some issues that need to be improved:
Q1:The abstract lacks a period at the end.
AU: We thank the reviewer´s comment. The period has been added.
Q2:L130, there should be a space between the number and the unit, and the whole text should be modified in turn.
AU: We appreciate the reviewer’s contribution. The text has been modified as suggested by the reviewer to correct the imprecisions previously written.
Q3:L186,the word “study” should be changed to “studies”.
AU: We thank the reviewer´s comment. In this case, we use the word 'study' in the singular to refer specifically to the study mentioned above.
Q4:L213, Table 1 headings should be centered, full text modification.
AU: The manuscript has been reviewed and corrected following the given observations.
Q5:L213, the word “0,45” should be changed to “0.45”.
AU: We thank the reviewer´s comment. The number has been modified.
Q6:L213,the Phenolic acids in Table 1 should conform to the format of Coumarins above, and be modified in the following paragraphs.
AU: This is a general section, so it is different.
Q7:L288, the format in Table 2 remains consistent,and the word “temperature gradient(75-60°)” should be changed to “temperature gradient(75-60℃)”.
AU: We thank the reviewer´s comment. The word has been modified.
Q8:L331-332, vitamin k related description is too little, please add.
AU: AU: We appreciate the reviewer’s contribution. The information on vitamin K has been expanded.
Q9:L400, data sink error occurred in the Carbohydrates section in Table 2.
AU: Table 2 is referred to Glucosinolates and their breakdown products present in broccoli by-products.
Q10:L488-490, this paragraph does not connect with the context.
AU: The paragraph has been modified as suggested by the reviewer to connect with the context.
Q11:L611, the format of the Major results section in Table 3 is consistent, all of which are in full-head or indented form.
AU: We thank to the review for the comment.
Q12:L686-687, the location of reference 29 is incorrect, and “incorporated” is not capitalized at the beginning of a sentence.
AU: According to reviewer comment, the reference has been modified.
Q13:Information in references should be complete and formatted consistently.
Please revise them.
AU: We appreciate the reviewer's comment. The information in the references has been completed according to the following format.

Reviewer 5 Report
Comments and Suggestions for Authors
This review study analyzes the bioactive chemicals found in broccoli by-products, focusing on their functional qualities and prospective applications. It focuses on key chemicals including glucosinolates, phenolic acids, flavonoids, organic acids, and pigments and discusses their health benefits. The manuscript also discusses several extraction and analytical methods for increasing the yield of these bioactive molecules, emphasizing the significance of selecting optimal extraction methods based on the target compounds.
The manuscript is well-structured. All of the major studies are addressed, and I applaud the authors for their tremendous effort. I offer three suggestions to improve the manuscripts, but even without them, it is suitable for publishing.
1) While the research discusses the overall health benefits of broccoli byproducts, it does not go into detail about clinical applications or specific effects on human health. Further research on bioactive chemicals' roles in clinical settings or direct health benefits would provide a more complete grasp of their practical applications.
2) In addition, the review might go into further detail about suggested paths for future research, including the potential of these chemicals in therapeutic or preventive health uses. Discussion of present knowledge gaps, such as interactions between bioactive chemicals and other dietary components or processing stability concerns, might aid in the direction of future research.
3) While the paper discusses the environmental benefits of employing byproducts, it might incorporate more practical solutions for implementing sustainable practices in agricultural and industrial settings.
Author Response
Revisions made to the manuscript are highlighted in red so that editors and reviewers can see the changes made. Three reviewers have already replied to the manuscript.
Reviewer 5
This review study analyzes the bioactive chemicals found in broccoli by-products, focusing on their functional qualities and prospective applications. It focuses on key chemicals including glucosinolates, phenolic acids, flavonoids, organic acids, and pigments and discusses their health benefits. The manuscript also discusses several extraction and analytical methods for increasing the yield of these bioactive molecules, emphasizing the significance of selecting optimal extraction methods based on the target compounds.
AU: We thank and appreciate the reviewer for all the comments.
The manuscript is well-structured. All of the major studies are addressed, and I applaud the authors for their tremendous effort. I offer three suggestions to improve the manuscripts, but even without them, it is suitable for publishing.
- While the research discusses the overall health benefits of broccoli byproducts, it does not go into detail about clinical applications or specific effects on human health. Further research on bioactive chemicals' roles in clinical settings or direct health benefits would provide a more complete grasp of their practical applications.
AU: We thank the reviewer´s comment. You are right that although the research is focused on general health benefits, there is still a need for further, specific clinical studies on the effects of broccoli by-products on human health. This review is already quite extensive, so including that line of analysis was not possible on this occasion. However, we will consider your suggestion for a future line of research that addresses in more detail the specific effects on human health and possible clinical applications of these bioactive substances.
- In addition, the review might go into further detail about suggested paths for future research, including the potential of these chemicals in therapeutic or preventive health uses. Discussion of present knowledge gaps, such as interactions between bioactive chemicals and other dietary components or processing stability concerns, might aid in the direction of future research.
We appreciate the reviewer’s suggestions in this point. We recognize that further exploring avenues for future research, especially around the therapeutic and preventative potential of these substances, would be important. We also agree that addressing knowledge gaps, such as interactions with other dietary components and concerns about stability during processing, would provide a more comprehensive approach for future research. We will take these recommendations into account to guide future reviews and studies, in order to contribute to a more detailed and applied understanding of these bioactive compounds.
- While the paper discusses the environmental benefits of employing byproducts, it might incorporate more practical solutions for implementing sustainable practices in agricultural and industrial settings.
We appreciate the reviewer´s comment. The authors agree that it would be valuable to include practical solutions to implement sustainable practices in agricultural and industrial settings. However, due to the scope of the work and the amount of information collected, it was not possible to address it in this review. We will consider your suggestion to explore these strategies in future works, in order to provide concrete and applicable solutions in these contexts.

Round 2
Reviewer 3 Report
Comments and Suggestions for Authors
This paper has reviewed analysis of nutritionally valuable compounds in various parts of broccoli based on processing methods and analytical techniques. Furthermore, this paper reviewed beneficial effects of broccoli and also various applications. This paper seems to be informative and could be valuable for the readers. Especially some tables are well summarized and very informative.
However, several things should be corrected or be reconsidered.
1. In the section of 3.8. lipids, authors seemed to be confused between lipids and fatty acids. In line 375, fatty acids should be changed to lipids.
2. Review paper should not just transfer data from articles. In the several cases, it is necessary to modify the data for comparison. For example, in case of total chlorophyll in table 3, mg/g DW, mg/100 g DW, μg/g DW were used. These data could be converted easily in the same unit, and then readers can compare the results easily. In case of Na, mg/100 g DW, mg/g DW, and mg/kg DW were used. These data should be converted into the same unit. In case of total protein in table 4, %(W/W) DW, g/100 g DW were used. In fact, they are the same unit. This should be corrected. I just showed some examples. The others like these examples should be corrected.
3. In the section 3.7.1 amino acids, amino acids should be written in free amino acids. However, the amino acid data in table 4 do not give any information. It may be better to remove this section.
4. In the section 3.9.2. sugars, contents are very confusing. Do sugars mean free sugars or sugars including polysaccharides? It is necessary to explain in detail including experimental methods if necessary.
5. In the section of beneficial effects, it will be better to include effects of broccoli on gut health and gut microbiome because gut health is quite important and many reports related to the effects of broccoli on gut health have been published. Furthermore, recently effects of phytochemicals on gut health become new area for investigation.
Author Response
Reviewer 3
Comments and Suggestions for Authors
This paper has reviewed analysis of nutritionally valuable compounds in various parts of broccoli based on processing methods and analytical techniques. Furthermore, this paper reviewed beneficial effects of broccoli and also various applications. This paper seems to be informative and could be valuable for the readers. Especially some tables are well summarized and very informative.
However, several things should be corrected or be reconsidered.
- In the section of 3.8. lipids, authors seemed to be confused between lipids and fatty acids. In line 375, fatty acids should be changed to lipids.
AU: According to reviewer comment, the redaction has been modified to clarify the meaning of the sentence.
- Review paper should not just transfer data from articles. In the several cases, it is necessary to modify the data for comparison. For example, in case of total chlorophyll in table 3, mg/g DW, mg/100 g DW, μg/g DW were used. These data could be converted easily in the same unit, and then readers can compare the results easily. In case of Na, mg/100 g DW, mg/g DW, and mg/kg DW were used. These data should be converted into the same unit. In case of total protein in table 4, %(W/W) DW, g/100 g DW were used. In fact, they are the same unit. This should be corrected. I just showed some examples. The others like these examples should be corrected.
AU: We appreciate your observation regarding the unification of units to facilitate the comparison of data across the studies presented in the tables. We agree that this approach could make direct comparisons between articles easier. However, as this is a review paper, we believe it is more appropriate to keep the data in the original units reported by the authors in the cited studies. This ensures the integrity and fidelity of the information as it was presented in the primary sources.
- In the section 3.7.1 amino acids, amino acids should be written in free amino acids. However, the amino acid data in table 4 do not give any information. It may be better to remove this section.
AU: Thank you for your comment. We consider it necessary to include the information on free amino acids in section 3.7.1, since this is a comprehensive review of the studies on the compounds present in broccoli by-products. Table 4 provides a summary of the data available in the literature on amino acids, which, although does not provide new information, is essential to provide a complete overview of the compounds studied in this context. We believe that maintaining this section contributes to the review's objective of providing a comprehensive and detailed review.
- In the section 3.9.2. sugars, contents are very confusing. Do sugars mean free sugars or sugars including polysaccharides? It is necessary to explain in detail including experimental methods if necessary.
AU: We appreciate your comment and realize that section 3.9.2, on sugars, can be confusing. To clarify, in this study we refer to “sugars” as free sugars, which includes monosaccharides and disaccharides present in the samples. We have revised the wording to make this distinction clearer and included experimental methods, which focus on the quantification of these specific sugars.
- In the section of beneficial effects, it will be better to include effects of broccoli on gut health and gut microbiome because gut health is quite important and many reports related to the effects of broccoli on gut health have been published. Furthermore, recently effects of phytochemicals on gut health become new area for investigation.
AU: We appreciate the reviewer’s contribution. A new section has been added to section 4 on the beneficial effect on intestinal health.
